

# Three-decadal dynamics of mid-channel bars in downstream of the Three Gorges Dam, China

Zhaofei Wen[1,2], Hong Yang[3,4], Changhong Ding[5], Ce Zhang[6], Guofan Shao[2], Jilong Chen[1], Mingquan Lv[1], Shengjun Wu[1], Zhenfeng Shao[7]

[1]Key Laboratory of Reservoir Aquatic Environment, Chongqing Institute of Green and Intelligent Technology, Chinese Academy of Sciences, Chongqing, 400714, China
[2]Department of Forestry and Natural Resources, Purdue University, West Lafayette, 47906, USA
[3]State Key Laboratory of Lake Science and Environment, Nanjing Institute of Geography and Limnology, Chinese Academy of Sciences, Beijing East Road 73, Nanjing, 210008, China
[4]epartment of Geography and Environmental Science, University of Reading, Whiteknights, Reading RG6 6AB, UK
[5]Aviation University Air Force, Changchun, 130022, China
[6]Lancaster Environment Centre, Lancaster University, Lancaster, LA1 4YQ, UK
[7]State Key Laboratory of Information Engineering in Surveying, Mapping and Remote Sensing, Wuhan University, Wuhan, 430079, China

*Correspondence to*: Zhaofei Wen (wenzhaofei@cigit.ac.cn)

**Abstract.** The downstream of the world's largest Three Gorges Dam (TGD) along Yangtze River (1560 km) hosts numerous mid-channel bars (MCBs). The morphodynamics of these MCBs are crucial to the river's hydrological processing, local ecological functioning and socioeconomic development. However, a systematic understanding of such dynamics and their linkages to the TGD across the entire downstream reach remain largely unknown. Using Landsat archive images and a thematic extracting scheme, the work presents comprehensive monitoring of MCB dynamics in terms of number, area, and shape, in the downstream of the TGD during 1985–2018. Total 140 MCBs were extracted and grouped into four types to represent small size ($< 2$ km$^2$), middle size (2 km2 - 7 km$^2$), large size (7 km2 - 33 km$^2$) and extra-large size ($>33$ km$^2$) MCBs, respectively. Most of the MCBs in terms of number (70%) and total area (91%) were concentrated in the lower reach (Hankou to Estuary). The number of small size MCBs decreased after TGD closure and most of such decreasing events happened in the lower reach. Although all four types of MCBs showed overall increasing trends in area, small MCBs had smaller rate and received more influence by the TGD operation than the large MCBs. Small size MCBs tended to become relatively shorter and wider whereas the others became slimmer after TGD operation. Impacts from the TGD operation could decrease along with the increasing distance from TGD to Hankou (for shape dynamics) or to Jiujiang (for area dynamics). The quantified longitudinal and temporal dynamics of MCBs across the whole downstream of the TGD provides a crucial monitoring basis for continuous investigations of changing mechanisms in the morphology of the Yangtze River system.



# 1 Introduction

Most of large rivers in the world are strongly regulated and engineered by artificial facilities (Freyer and Jefferson, 2013). A fundamental type of these facilities are dams, as they lead to discontinuity and alteration in the hydrological and ecological processes of river systems (Hecht et al., 2019; Yang and Flower, 2012). Accordingly, increasing attention has been paid to
the various impacts of damming, especially in regard to the downstream channel morphodynamics and to its hydrological and ecological processes (Liermann et al., 2012; Schmutz and Moog, 2018).

One of the major geomorphic features that are potentially affected by damming is the mid-channel bar (MCB), or named as river island (i.e., elevated region that surrounded by river channel) (Osterkamp et al., 1998; Adami et al., 2016). As an essential geomorphic unit and component in fluvial system, MCBs dynamics directly influences river's morphological
evolution and sediment archiving through their interaction with the flow and sediment transport (Ashworth, 1996; Hooke, 1986; Xu, 1997; Wintenberger et al., 2015). Therefore, MCBs dynamics under the context of damming intervention are considered as special importance in maintaining the stability of river channels (Lou et al., 2018; Sanford, 2007) and ecosystem functions (e.g., biodiversity and habitat providing) of the related hydro-ecosystems, including the island ecosystem, adjacent riverbed and wetland (Tonkin et al., 2018; Tabacchi et al., 2009).

MCBs cannot often remain stable. They, instead, experience constant changes in terms of area, shape, height and location, due to the longitudinal and temporal varying conditions of the flow and sediment load (Wang, 2017; Hooke and Yorke, 2011). With intensive human interventions like dam operation, the natural river regimes are gradually replaced by the regulated flow and sediment load, which dramatically affected the MCBs' evolution and thereby gaining scientific and social concerns internationally (Wintenberger et al., 2015; Han et al., 2017; Nelson et al., 2013). For example, the dynamics of
MCBs together with the change in dam-induced channel hydrologic regime have been evidenced in large rivers across the globe, such as the UK (Petts, 1979), Australia (Erskine, 1985), the USA (Graf, 2006), France (Provansal et al., 2014) and China (Wang et al., 2018a).

Generally, the dynamic behavior of single MCB in short term tends to be less informative (Xu, 1997). When considering the long-term and long-reach river at a given frequency of discharge, the MCBs, however, might illustrate some statistical
characteristics of dynamic behaviors, and thus remarkable variations between different MCBs can be observed (Lou et al., 2018; Wintenberger et al., 2015). Besides, it is possible to explore the longitudinal and temporal dynamics of MCBs, to reveal the response of bar development to the associated impact factors (e.g., damming, water flow, sediment load, and boundary conditions), and eventually, to project the MCB evolution in the future scenarios (Xu, 1997; Dai and Liu, 2013). Therefore, the issue of MCB dynamics is significantly important not only in geomorphic theory, but also in hydraulic
engineering practice, particularly in both ecological and economical importance of large rivers that are under extensive anthropogenic influences, has appeared to be a pressing need (Wintenberger et al., 2015; Adami et al., 2016).

As the largest river in China, the Yangtze River plays a key role in maintaining ecological security (e.g., wetland protection, fishing management and shoreline protection) and economy development (e.g., shipping, sand supply and land



development) (Xu et al., 2006). This is especially the case for the downstream reach of the three gorges dam (TGD) on the Yangtze River as it passes through one of the country's most densely populated and most industrialized region (Yang et al., 2012; Zheng et al., 2018). The survey in very early 1980s found hundreds of MCBs with a total area over 1000 $km^2$ in the downstream of the TGD (downstream of Yichang) (Sun, 1984). These MCBs are often considered as important land

resources for channel dredging, habitat providing, and even agricultural development and urbanization, depending on the areas and geographical conditions (e.g., stability, vegetation and soil) (Lou et al., 2018). Over the past decades, the hydrodynamics in this reach, however, experienced huge changes due to various anthropogenic activities, and the most significant influence is the construction of the world's largest dam TGD (Gao et al., 2013a; Hu and Hou, 2015). Under such environmental alterations and by considering the ecological and economical importance of the MCBs, the most intuitive

questions have been raised: (1) How do the dynamics of MCBs act pre- and post- the damming periods? (2) To what extent are they influenced by the damming?

The investigations of the MCB dynamics with respects to the damming or reservoir construction have last over five decades. Petts (1979) reviewed potential bar dynamics subsequent to dam and reservoir construction along multiple British rivers. The earlier investigations were focused more on the geomorphologic theory, such as description of development

stages (e.g., formation, migration, translation and channel equilibrium) through experimental work or field work but within a limited longitudinal-temporal span (Hooke, 1986). Since 1980s, researches on MCBs dynamics in the context of long reach and long-term temporal span, however, are paid increasingly wider attentions (Hooke and Yorke, 2011; Wintenberger et al., 2015), thanks to the rapid development of Remote Sensing (RS) technology and the sophisticated requirements for river regulating (Baki and Gan, 2012; Gao et al., 2013b). Among these studies, the common observations are that the area and

number of MCBs tend to decrease due to the dam-induced sediment supply reducing (Freyer and Jefferson, 2013; Wang et al., 2018a; Raška et al., 2017; Lou et al., 2018). In other cases, however, different dynamical patterns have been observed. Sanford (2007) showed that in the Missouri River, dam regulated rate of flow with the consequence of reduce in the bars' area, less centroid migration but more easy bars aggregation. Kiss and Balogh (2015) showed the MCBs developed quickly upstream and laterally in the whole Dráva River, because of the coarse sediment supply and the decreasing stream energy by

the dam effect. This situation suggests that the downstream geomorphic equilibration to dam operation could result in MCBs emergence, stability, or erosion, depending upon the new hydraulic regime, sediment supply, and the type of structures employed (Piégay et al., 2009; Freyer and Jefferson, 2013).

Since the closure of the TGD, the effects of dam operation on the dam's downstream have been heavily discussed. Most of the topics are focused on the changes in flow and sediment regime and channel morphology (Han et al., 2017; Lyu et al.,

2019; Yang et al., 2011). However, only few works are conducted on the MCBs dynamics using time-series RS images (Gao et al., 2013b; Wang et al., 2018a; Lou et al., 2018; Wang et al., 2018c). For example, in the immediate downstream of the TGD, the total area of MCBs extracted from Landsat images reduces drastically by 19.23% from 2003 to 2016, accompanied with an increase in water surface width (Wang et al., 2018a). Lou et al. (2018) focused on the lower reach, MCBs' total area exhibited a decrease in pre-TGD period, while an increase in the post-TGD period. It is noted that these results are focused



on specific reach and on only the big MCBs with only concern about area metric. Such rare information is hardly to support a systemic understanding of the MCB dynamics over the entire downstream of the TGD.

To better understand the above concerns, our study made the attempt to comprehensively analyze the MCBs dynamics over the entire downstream of TGD throughout the last three decades (covering pre-TGD and post-TGD periods), by (1) inventorying explicit MCBs extracted from time-series satellite imagery and (2) exploring the spatial (or longitudinal) and temporal patterns of the MCBs on the local damming and other anthropogenic contexts with three morphology-related metrics (i.e., number, area and shape). Following the introduction, Section 2 describes the study area, data sources including satellite images and gauging station data, and methods for processing and analyzing image data. Section 3 presents the accuracy results of MCBs and the longitudinal-temporal dynamics of MCBs. Section 4 discusses the scale effects and the driving factors of such dynamics. Conclusions are drawn in the end.

## 2 Material and Methods

### 2.1 Study area

Yangtze River (or Changjiang in Chinese), originates on the Qinghai-Tibet Plateau 5100 m above sea level, and flows eastward to the East China Sea. It is the largest river on the Eurasian continent and third longest river in the world, with a length of 6300 km and a drainage area of about $1.8 \times 10^6$ km$^2$. The climate is controlled by the East Asian Monsoon with the basin-wide precipitation averages around 1100 mm/yr, and approximately half of which is lost to evaporation(Han et al., 2015). Geographically, the mainstream of the Yangtze River consists of the three reaches (Fig. 1): the upper reach, the upstream of YC (Yichang), the middle reach from YC to HK (Hankou), and the lower reach from HK to the estuary (EST) near Shanghai. The TGD lies in 43 km upstream of YC (Fig. 1). It is the largest hydropower dam around worldwide. The TGD was closed in 2003 and fully operated after 2010. The principle operation of the TGD is to keep the behind Three Gorges Reservoir (TGR) at the lowest water level of 145 m in flooding season from May to September and at the highest water level of 175 m in dry season from October to April in the following year (Yang et al., 2014). Accordingly, the hydrological and sediment regimes in the downstream of the TGD are also altered (Dai and Liu, 2013).

The downstream reach of the TGD is served as our study area (Fig. 1). It is 1560 km long and consists of both middle and lower reaches of the Yangtze River. The reach is much wider than the upper stream and the riverbed slope is generally small. Around the reach is plain area which is considered as one of China's most densely populated and highly industrialized area (especially in the lower part) (Yang et al., 2014). Thus, apart from hydrology alteration by the TGD operation, it is also affected by other human activities, such as navigation, sand mining, agricultural activates, and urbanizations. What's more, the ecological functioning values of this reach are highly recognized, such as wetland protection, rare animals' protection, and ecological sustainability maintenance (Liu et al., 2008).



## 2.2 Data sources

### 2.2.1 Landsat images

The free Landsat archive images obtained during the period of 1985 - 2018 were used as raw image source for extracting long-term MCBs datasets. They include Landsat -4/5 TM (1985-2011), -7 ETM+ SLC-ON (1999-2003), -7 ETM+ SLC-OFF (2009-2012), and -8 OLI images (2013-2017). All the images are standard Landsat surface reflectance level-2 products in 30m spatial resolution. They were ordered from the United States Geological Survey (USGS)'s Earth Explorer Website (http://earthexplorer.usgs.gov/), and downloaded from the USGS Earth Resources Observation and Science Center Science Processing Architecture on Demand Interface (https://espa.cr.usgs.gov/). Landsat-4/5 TM or Landsat-7 ETM+ surface reflectance data were produced by the software of Landsat Ecosystem Disturbance Adaptive Processing System (LEDAPS) (Masek et al., 2006). More details about the LEDAPS algorithm and Landsat-4/5/7 Surface Reflectance data products can be found in the Product Guide (2018b). Landsat-8 surface reflectance data were generated from the Landsat Surface Reflectance Code (LaSRC), which makes use of the coastal aerosol band to perform aerosol inversion tests, uses auxiliary climate data from MODIS and uses a unique radiative transfer model. Specific details about the LaSRC algorithm and Landsat 8 Surface Reflectance data products can be found in Product Guide (2018a). Moreover, for a specific scene (e.g., path = 121, row = 039), the images are consistently georegistered within prescribed image-to-image tolerances of less than 12 m, making them suitable for time-series pixel-level analysis (Young et al., 2017).

Not all Landsat images covering the study area are suitable for the analysis. To minimum the uncertainty in the MCBs' extracting processing, two criteria must be met in screening images. First, the cloud cover should be less than 10% of the whole scene since it is considered as negative information in extracting works. Second, images should be acquired during the dry season of the study area (i.e., November to March) (Dai and Liu, 2013). The reasons are given here. First, the presence of MCBs is largely affected by the water level at the imaging time. Thus, it is important to make sure the water levels are in exactly consistence among different imaging times for the temporal analysis of MCBs. However, such ideal requirement is unrealistic for the Landsat images acquired at frequency of 16 days, and the practical way is to keep the water levels are in relatively less variations. Second, the water level variation in dry season is much smaller than in the flood season (Fig. 2), which may minimum the uncertain of MCBs' presence caused by water level fluctuation (Wang et al., 2013). Finally, 553 images were filtered out for extracting MCBs (see Supplementary Table S1).

### 2.2.2 High spatial resolution (HSR) images

Through Google Earth images, some HSR images were carefully selected as sources data to obtain the validation of MCBs, which were further applied to assess the accuracy of MCBs that are extracted from the Landsat images. Since the present of MCBs is highly dependent on the water level that is temporally varied, the acquisition dates of one Landsat image and the corresponding HSR image should be the same. However, this kind of ideal criterion cannot always meet in practice. To obtain enough validation MCBs for accuracy assessment, we reduced the criterion to that the acquisition date of HSR image





should be within 7 days from Landsat image. Finally, 19 HRS images were selected and downloaded from Google Earth (Table 1).

### 2.2.3 Gauged datasets

Water flow and suspended sediment discharges were collected at the three key gauging stations (YC, HK, and DT) from annually released Yangtze River Sediment Bulletin (http://www.cjw.gov.cn/zwzc/bmgb/). These data were yearly aggregated from daily measurements. The measurements of daily water flow and suspended sediment discharges following six standard steps: (1) collecting 10–30 vertical profiles for measurement and the final profiles number is determined by river width; (2) measuring flow velocities (using a velocimeter) at six depths vertically located from water surface to riverbed with vertical interval of 0.2 H, where H is the depth of the water column; (3) after measuring the flow velocities, sampling water with horizontally-oriented water bottles at the same depth; (4) drying the water samples to determine suspended sediment concentrations (SSCs); (5) calculating water discharge using the product of the cross-section area of that station and mean flow velocity; and (6) calculating sediment discharge using the product of water discharge and SSC. (Ministry of Water Resources the People's Republic of China, 1992; Yang et al., 2014).

### 2.3 MCBs producing framework

### 2.3.1 Automatic extracting MCBs

The key idea of automatic extracting MCBs is to obtain the river's water body from Landsat images at first. Then, the holes distributed in the water body are therefore considered as MCBs as they are surrounded by river water in the real world. The processing consists of five main steps as illustrated in Fig. 3 and detailed below.

*Step 1: Image clipping*

Compared to a whole Landsat scene, the area occupied by the Yangtze River channel is relatively small. To improve the compute efficiency, it is necessary to clip the small Yangtze River channel area from the Landsat scene for further processing. The clipper file was a 1-km buffered Yangtze River polygon GIS data which was manually delineated from Landsat-8 images acquired in the dry season from 2015 to 2017 over the entire study area (Fig. 1).

*Step 2: Calculating remotely sensed water index*

Remotely sensed water indexes have widely been used for mapping surface water bodies and better understanding of the water resource change (Verpoorter et al., 2014). Many of them were designed originally for Landsat imageries and have shown excellent performances (Fisher et al., 2016). For example, the normalized different water index (NDWI) (Mcfeeters, 1996), the modified normalized different water index (MNDWI) (Xu, 2006), the automated water extraction index for




images with shadows and without shadows (Feyisa et al., 2014), and the water indices developed for Australia in 2006 (Danaher and Collett, 2006) and 2015 (Fisher et al., 2016). In this study, the MNDWI was adopted since the algorithm (Eq. 1) is less sensitive to specific sensors compared with others (except for the NDWI) (Xu, 2006). Thus the MNDWI could be suitable for all the Landsat-4/5/7/8 images. In addition, the overall performance of MNDWI is shown outstanding through

comparative analysis in previous studies (Ji et al., 2009; Zhang et al., 2018; Fisher et al., 2016; Feyisa et al., 2014). Eq. (1) was applied to all the clipped satellite images to generate MNDWI image by Python script in ArcGIS environment (version 10.3.0.4322) (ESRI, 2014).

$$\mathrm{MNDWI} = \frac{\rho_{\mathrm{green}} - \rho_{\mathrm{swir1}}}{\rho_{\mathrm{green}} + \rho_{\mathrm{swir1}}} \tag{1}$$

where $\rho_{\mathrm{green}}$ and $\rho_{\mathrm{swir1}}$ are surface reflectance in green band and short-wave infrared band (i.e., band 5 for Landsat-4/5/7

and band 6 for Landsat-8), respectively.

*Step 3: Determining optimal threshold*

To classify river water body from land area, an optimal threshold is needed to apply on the MNDWI image. However, the optimal threshold often varies with space and time (Ji et al., 2009). Because of hundreds of MNDWI image, it is crucial to determine the optimal threshold of each MNDWI image automatically. The modified histogram bimodal method (MHBM)

developed by Zhang et al. (2018) was applied to automate determine dynamic threshold in the study.

*Step 4: Classifying water and non-water areas*

For a MNDWI image, pixels with values larger than the determined threshold were classified as water area the others were classified as non-water area.

*Step 5: Extracting MCBs*

Firstly, we converted the raster format water body area into vector shapefile polygons with smooth edges. Then the holes in the vector water body were converted to polygons which were considered as candidate MCBs. Geographically, all the holes contained in river water body are MCBs. For the Yangtze River, however, some of the holes can also be large ships because this river is one of busiest inland waterways in the world (Zhang et al., 2014). Such detected ships are often less than 0.02 km$^2$ after carefully inspected by the first author on the high spatial resolution images provided by the Google Earth. Finally,

the "true" MCBs were selected from the candidate MCBs with area larger than 0.02 km$^2$. All these processes were done by specific Python scripts in ArcGIS environment (version 10.3.0.4322) (ESRI, 2014).



### 2.3.2 Calculating MCBs' attributions

*(1) Unique Coding*

Each MCB generated from a Landsat scene was initially assigned a unique identification code. The code consists of three parts: the scene's path and row numbers, the scene's acquired date, and the MCB's geometry identification which was

automatically generated by the ArcMap Desktop (version 10.3.0.4322) (ESRI, 2014). For example, code "12103920100114005" means the corresponding MCB is identified as "005" in that scene which was acquired at path "121" and row "039" on date January 14, 2010. However, it worth noting that the same MCBs generated from different Landsat scenes are with different identification codes, which make the temporal variation analysis of individual MCBs difficultly. Thus, the MCBs which located in the same place but generated from different Landsat scenes with different acquisition dates

should share the same code. In this study, the code which contains the earliest acquisition date was assigned to all the MCBs with same location.

*(2) Area and shape index*

In our analysis, the area and shape attribution were focused. Since MCB often presents in transverse or lobate shape (Hooke, 1986), the length width ratio (LWR, Eq. (2)) may serve as one of good shape indexes to indicate the MCB's shape

characteristic and its dynamics (causing by deposition or erosion).

$$LWR = \frac{L}{W} \tag{2}$$

where *L* and *W* represent length and width of convex rectangle of the MCB, respectively. Both MCB's area and convex rectangle were generated with ArcMap Desktop (version 10.3.0.4322) (ESRI, 2014).

### 2.4 Validation data processing

The validation MCBs that present in the HSR images (Table 1) were manually delineated with ArcMap Desktop environment (version 10.3.0.4322). At the same time, the corresponding assessment MCBs generated by Landsat images (Table 1) were also picked out. In total, 49 pairs of MCBs (validation MCBs VS. assessment MCBs) that are matched (see Supplementary kml files). Both their area and LWR value were calculated and were used for further accuracy assessment analysis.

### 25   2.5 Analysis methods

A time series of environment related data can sometime contain a structural break, due to sudden disaster or human intervention. Previous case studies have reported that the temporal dynamics of some MCBs may be changed by the closure of the TGD (Gao et al., 2013b; Lou et al., 2018). In our study, this kind of structural breaks were tested systematically. The Chow test (1960) used by many other researchers (Lewis and Landry, 2017; Lee, 2010) was applied to test the TGD induced



structural break. The Chow test of time series MCB data examines two hypotheses (H0: there is no structural change; and H1: there is a structural change). It is conducted by running three separate regressions and an *F*-test: (1) a linear regression with the entire time series data; (2) two separate linear regressions with data before (Pre-TGD) and after TGD closure in 2003 (Post-TGD); and (3) an *F*-test is used to determine whether a single linear regression is more efficient than two

separate regressions. If this *F*-test was significant (*p*-value < 0.05), the null hypothesis (H0) was rejected and the alternative hypothesis (H1) was accepted. This process was applied on each individual MCB and grouped MCBs (as detailed in Section 3.2) with R environment (R Core Team, 2013).

## 3 Results

### 3.1 Accuracy of MCBs

Accuracy of the MCBs that extracted from Landsat images were assessed by the validation MCBs that generated from HSR images. It is noteworthy that the validation data were representative from the perspectives of spatial distribution (scattering the entire study area as listed in Table 1), size of MCBs (covering a wide range of areas as shown in Fig. 3a), and shape characteristic (including a wide range of LWR index as shown in Fig. 3b). For both area and LWR, the Landsat induced MCBs were largely consistence with the validation MCBs. The RMSE values for area and LRW were 0.25 km$^2$ and 0.22,

respectively. The result shows that the framework proposed for extracting MCBs was reliable and the accuracy of Landsat image based MCBs was high enough and suitable for further analysis of area and LWR dynamics.

### 3.2 Basic statistics of MCBs

This study provides us the first opportunity for knowing general picture of MCBs in the downstream of the TGD. Area of individual MCB varied from the smallest of 0.08 km$^2$ to the largest of 223 km$^2$ (Fig.5). Moreover, the area histogram

illustrates a strong right-skewed distribution as shown in Figs. 5a-1 and 5a-2. Specifically, MCBs with areas of < 2 km$^2$ contributed 50% to the total MCBs; MCBs with areas of 2 km$^2$ - 7 km$^2$ accounted for 25%; MCBs with area 7 km$^2$ - 33 km$^2$ contributed 20%; and MCBs with area > 33 km$^2$ only counted for 5%.

Area is a comprehensive indicator of both geographical processes (e.g., geomorphology evolution stage, vegetation stats, stability, and so on) and anthropological processes (e.g., land use type, developing history, sand mining, and so on)

(Ashworth, 1996; Hooke and Yorke, 2011). Thus, the observed huge area variations between MCBs indicate that the dynamics analysis of MCBs should fully consider the potential influences caused by different size or the scale effect. Classify all the MCBs based on a specific criteria is necessary. To the best of our knowledge, however, there are no such criteria that can be referenced. Most of previous studies focused on MCBs with specific size (i.e., relative limit area variation) (Gao et al., 2013b; Lou et al., 2018). In this study, considering both the area histogram distribution pattern (Figs. a-1 and a-2)

and sample size for further statistical requirement, MCBs were grouped into four types, namely T1, T2, T3, and T4 (Table 2).



### 3.3 Longitudinal distribution of MCBs

The longitudinal distribution of MCBs is illustrated in Fig. 5c. As expected, the distribution pattern is uneven. Overall, the longer distance to the TGD, the more density of MCBs. Specifically, the distinct distribution patterns of MCBs in middle

reach (TGD-HK) and lower reach (HK-EST) were observed. In the middle reach, there were 42 MCBs from 1985 to 2018, account 30% of total MCBs. The average interval of these MCBs was 15 km and the total of their yearly averaged areas was 119 km$^2$. In the lower reach, there were  98 MCBs in the study period, over 2 times of those in the upper reach. Their presence density was also much higher (average interval =10 km) than that in the upper reach. Moreover, the total of their yearly averaged areas was 1172 km$^2$, almost 10 times of those in the upper reach. These patterns highlight the importance of

lower reach in the analysis of MCBs from both quality and quantity perspective of views.

To be more specifically, the MCBs in 2016 were taken as an example to illustrate the observation (Fig. 6). It is noteworthy that the T1 MCBs (i.e., small-size) scattered along the entire reach without obvious concentration. Most of the T2 and T3 MCBs (i.e., middle and large sizes) present in the lower reach. All of the T4 MCBs (i.e., extra-large size) are distributed in the lowest reach, i.e., JJ-EST (Fig. 6a). In other words, MCBs with large areas tend to be developed in the

lower of the downstream of the TGD, especially in the reach of JJ-EST.

### 3.4 Temporal dynamics of MCBs

The temporal dynamics of MCBs in terms of total number, area, and LRW indexes are addressed below. For each index, with and without consideration of MCB types are both involved. In each temporal trend analysis, whether the closure of the TGD can be considered as a structural change during the whole trend is also tested. If it does, the pre-TGD, post-TGD, and

the overall regressions are applied; otherwise, only the overall regression is applied.

### 3.4.1 MCB number

For the total MCBs, there was no statistically significant change in MCB number during the whole study period as shown in Fig.7a ($a$ = -0.08, $p$-value > 0.05). However, the TGD closure year (in 2003) was detected as a structural change in the whole period and different trends were observed between the pre-TGD and post-TGD periods. In the pre-TGD period, MCB

number showed an increasing trend ($a$ = 0.4 and $p$-value < 0.01). Such trend was reversed in the post-TGD period in which a significant decreasing trend was detected ($a$ = -0.5, $p$-value < 0.01). To further understand the these changes, trends in the four MCB types were checked as illustrated in Figs. 7b, 7c, 7d and 7e. The number of T1 MCBs showed very similar trends as the total MCBs and the number in other three MCB types remained stable during the study period, suggesting that the phenomenon of MCB's "disappearance" or "appearance" existed in the study periods and only applied in the small size

MCBs (T1). It should be noted that the so-called "disappearance" or "appearance" did not mean a MCB just disappeared to nothing or appeared from nothing. This is because only the candidate MCBs with area larger than 0.02 km$^2$ were detected (as





explained in Section 2.3.1) and applied in our analysis. Therefore, in some cases, the "disappearance" of one MCB may be caused by flow erosion which reduced its area just under our detection criterion (0.02 km$^2$).

Since the number change only occurred in T1 MCBs, where and when those events happened is another question. As shown in Fig. 8, the appearances of new MCBs mainly happened in the pre-TGD period and ended in 2004. The locations of those events were concentrated in the JJ-EST reach. The events of MCB disappearances started from 1996 and mainly located in the JJ-EST reach as well. It is unexpected that the immediate downstream reach of the TGD (i.e., TGD-HK) had relative little decrease in MCB numbers after the closure of TGD in 2003 (only 2 were spotted).

### 3.4.2 Area

Overall, the area of total MCBs experienced an increasing trend with the slope of 2.77 km$^2$/yr ($p$-value < 0.01, Fig. 9). Area of all four MCBs types presented increasing trends (trend of the T3 is not statistically significant). The slopes of T1, T2, and T4 increased accordingly (Fig. 9), from 0.53 km$^2$/yr ($p$-value < 0.01) to 0.61 km$^2$/yr ($p$-value < 0.01) to 1.29 km$^2$/yr ($p$-value < 0.01). In addition, it is also noteworthy the decreasing pattern of $CV$ values from T1 to T4. The combination of those observations may indicate that the larger size of MCBs tend to have higher rate in their area increasing (the more sediment deposition) but with less variability (or more stability).

As for the TGD effects on area dynamics, the structural changes were detected in T1 and T2 MCBs, but not in the T3 and T4 MCBs (Fig. 9). For T1 and T2 MCBs, their areas showed increasing trends in the pre-TGD period but showed slightly decreasing trends in the post-TGD. It suggests the closure of TGD might have significant impact on area dynamics of small (T1) and middle (T2) size MCBs. In other words, MCBs with small area were more likely to be influenced by the dam operation and got more sediment erosion whereas MCBs with large area probably had more resistance to the impact of dam operation.

### 3.4.3 LWR index

Different types of MCBs had distinct basic shape characteristics as shown in Fig. 10. A general decreasing trend of $\mu$ values was spotted with the increasing MCBs size, i.e., 4.97 (T1) > 4.12 > 3.16 (T3) > 2.24(T4). According to the definition of LWR, the observation shows that the MCBs with smaller area (e.g., T1 and T2) were likely to have slim shape, whereas MCBs with larger area (e.g., T3 and T4) were tend to be shaped in relative shorter and wider. Those shape characteristics were in accordance with the general evolution processing of MCBs (Hooke, 1986; Hooke and Yorke, 2011; Wintenberger et al., 2015).

As for the overall temporal dynamics, it is noted that the LWR of the total MCBs showed a significant decreasing trend in the whole study period ($a = $ -0.02, $p$-value < 0.01, Fig. 10a), suggesting that the shape of MCBs tended to change from slime to relative shorter and wider. But not all types of MCBs showed such similar shape trend. T1 and T2 MCBs showed the significant decreasing trends while T3 and T4 MCBs showed significant increasing trend in the whole period (Figs. 10b, 10c, 10d, and 10e). It means the MCBs with small and middle size tended to become relative shorter and wider whereas the





MCBs with large and extra-large size were tend to become slimmer. In spite of their change direction, their change rates were rather different with larger sizes tending to have smaller change rate, i.e., 0.03 /yr (T1) > 0.02 /yr (T2) > 0.01 /yr (T3) > 0.001 (T4). It indicates the shapes of T3 and T4 MCBs might be more stable than those of T1 and T2 MCBs.

Structural changes were also detected in the LWR dynamics of T1, T2, and T3 MCBs in 2003. For T1 MCBs (Fig. 10b), they experienced insignificant trend change in the pre-TGD period and significantly decreasing trend in the post-TGD period. Both T2 and T3 MCBs showed a significant increasing trend in the post-TGD period (Figs. 10c and 10d).

## 4 Discussion

### 4.1 Scale effects on the temporal dynamics

The temporal dynamics of MCBs indicate that different MCB types had distinct trend characteristics. More specifically, the larger MCBs seem to have more stability than the relative smaller ones in both area and LWR variation. In other words, the size of MCBs may influent their temporal dynamic patterns, which is also known as the scale effects. Fig. 11 illustrates the scale effects of MCBs on the temporal stabilities of area and LWR in the overall, pre-TGD, and post-TGD periods, respectively. It is clear that CV of area and of LWR were both decreasing (becoming less variation and more stability) with the increasing area or scale. Such pattern can apply for the overall, pre-TGD, and post-TGD periods. That is to say, the larger size of MCB, the relative larger stability in area and shape variations; whereas the smaller size of MCB, the relative less stability in area and shape variations. In the previous studies, only the specific scale of MCBs were focused (such as middle- and large-size MCBs), and the results may not be able to address the changes of small sizes MCBs (Wang et al., 2018a; Gao et al., 2013a).

Three reasons could explain the observed scale effect. The first one is the growth of vegetation. Compare to small MCBs, large MCBs are more likely grown vegetation (Fig. 12) (Asaeda and Rashid, 2012; Osterkamp and Hupp, 2010). In turn, the vegetations would provide more resistance to the external forces, such as sediment erosion (Gilvear and Willby, 2006; Hooke and Yorke, 2011). The second one is the strength of human intervenes. Large MCBs are more likely been intervened by human beings and received more protections which can enhance their stabilities. For example, people would harden MCBs' banks to prevent them from being eroded, especially on agriculturalized or industrized MCBs in the lower reach (Gregory, 2006). The third one is the flooding influences. Generally, the small MCBs have short evolution history with lower height compare to the large MCBs. Therefore, they are more likely been inundated in a flooding event and cause large change in sediment erosion or deposition (Wang et al., 2018c). On the contrary, large MCBs would receive relative fewer impacts by such flooding event (Fig. 12).

### 4.2 Potential effects of the TGD operation

It has been widely acknowledged that the presence of MCBs can be heavily influenced by the flow and sediment regimes (Raška et al., 2017; Gilvear and Willby, 2006). In the downstream of the TGD, the marked changes of flow and sediment



regimes between the pre- and post-TGD periods have been observed (Mei et al., 2015; Yang et al., 2014). Although many observations focused on different study periods, some common views can be draw here. First, there is no significant change in the annual water charge between pre-TGD and post-TGD periods in the last decades (Dai and Liu, 2013) (Fig. 12a). However, the intra-annual water flow pattern has been alternated by the operation of the TGD. Specifically, water charge in

flooding season has been reduced but it is increased in dry season (Lai et al., 2017). Second, the sediment amount carried by the water flow decreased from 1990s, but it dropped dramatically after TGD closure in 2003 (Fig. 10b) (Wang et al., 2018b). Third, the sediment carrying capacity decreased in the lower reach (Lai et al., 2017; Lou et al., 2018). The direct result of these changes is that the water flow in the immediate downstream of TGD is in a sediment-hungry condition which would carry away some sediment in riverbed or in the submerged parts of MCBs (Yang et al., 2015).

The operation of the TGD could have impacted on temporal dynamics of MCBs in the downstream reach. A further urged concern need to be discussed is that do these influences spatially vary or do the TGD effects vary against with the distance of MCBs to the TGD. Here the frequency of structural changes (FSC) is served as an indicator in analyzing that concern. The FSC is identified as the frequency of MCBs detected experiencing structural changes in 2003 among every 10 MCBs that are in an ascending sequence by their distances to the TGD. Therefore, the higher FSC, the greater chance of

such structural changes that being contributed by the TGD operation (Fig. 14a).

     For area temporal dynamics, the FSC gradually decreased from 0.7 to 0.2 in the TGD-JJ reach, and then increased gradually from 0.2 to 0.8 in the JJ-NJ reach (Fig. 14a). Similarly, the FSC of LWR gradually decreased from 0.7 to 0.3 in TGD-HK reach, and then increased gradually from 0.3 to 0.8 in the HK-DT reach. The decreasing patterns of FSC indicate that, to some extent, the effect of TGD operation decreased with the distance to the TGD, and the lowest effects occurred at

HK and JJ for LWR dynamics and area dynamics, respectively. This understanding is similar to the observations made in the analysis of TGD effects on the water regime and channel dynamics (Wang et al., 2013; Yuan et al., 2012). Moreover, it is noteworthy an unexpected increase trend of FSC in the HK-DT and JJ-NJ for LWR and area, respectively. Does it mean that the TGD effects increased in these lower reaches after the effects had been minimized as discussed just above? It may unreasonable to make such conclusion and thus more evidences are needed to explain the increasing pattern. In fact, apart

from the above mentioned hydrological factors, some human activities, such as sand mining, may play an important role in changing area and LWR dynamics of MCBs (Wang, 2017), especially for the middle- and small-size MCBs in the lower reach like JJ-DT. Before 2003, the sand mining activity was banned. However, after carrying out the Regulation of Sand Excavation Management in the Yangtze River in June 2003, the amount of yearly sand mining increased dramatically (Changjiang Water Resources Commission, 2003-2017). Those sand mining activities mainly focused on the lower reach,

particularly in the JJ-EST reach (Fig. 14b). Since the large scale of sand mining activities often happened near MCBs (Hu and Hou, 2015), it could cause structural change of MCBs dynamics in 2003. Therefore, the causes for MCBs' dynamics in the lower reach (JJ-EST) is more complicated than in the immediate downstream of the TGD (TGD-HK). Therefore, more supporting data and quantitative analysis are needed to more accurately evaluate the effects of TGD on that reach.





In addition to the longitudinal extent of TGD effects, we also like to know the major way in which the TGD effects played. As Fig. 15 illustrate, majority of structural changed MCBs experienced opposite trend change in post-TGD period compared to the pre-TGD period. Specifically, 72 out 140 MCBs showed PR structural change (positive in pre-TGD and negative in post-TGD) in area temporal dynamics and 79 out 140 MCBs experienced NR structural change (negative in pre-
TGD and positive in post-TGD) in LWR temporal dynamics. The observation suggests that the operation of the TGD may be the driving force that makes over half of MCBs experienced an erosion condition (decreasing area) and became slimmer (increasing LWR) than in pre-TGD. Thus the operation of TGD could have significant impacts on the dynamics of MCBs in the downstream of the TGD.

## 4.3 Limitation and future research

Similar as many studies, there are several limitations and open issues in the current research. The area and shape of image-extracted MCBs are influenced by the variation of water level on different image acquisition dates. Therefore, the temporal dynamic analyses of such MCBs may have a certain level of uncertainties due to the inconsistence of water levels on different image acquisition dates. This unwelcome factor has been carefully considered with our best effort in the study, while uncertainty may still remain in the analyses of area and LWR. Generally, two further improvements could be made to
minimize the impacts of such uncertainties. First, future studies can apply temporal dynamics analysis on a group of MCBs instead of individual ones and make conclusions from the statistics point of view (Dalvi and Suciu, 2005). Second, further researches can use newly high-temporal-resolution (1-3 day) RS images, such as PlanetScope (Planet Team, 2017), with higher consistency of the water levels on their acquisition dates. The second improvement is high recommended in the future MCBs monitoring works as these data can be obtained more economically in the near future.

Due to the relative coarse spatial resolution of Landsat images (30 m) and the complex conditions of Yangtze River, this study selected the area threshold of 0.02 km$^2$ and the image-extracted patches with area larger than such threshold were considered as MCBs. As a result, some MCBs with area less than 0.02 km$^2$ were excluded in the analysis. In the future study, however, these tiny MCBs can also be carefully manually investigated by sophistical experts or by using high spatial resolution images to separate them from non-MCBs objects, such as large ships and some man-made features.

The MCB dynamics not only perform as area and shape change in 2-D space, but also act as volume change and migrating in 3-D dimension (Wang, 2017; Adami et al., 2016). To fully understand the dynamics of MCBs and the impacts of TGD, more future works needs to be conducted in monitoring the migration and sand volume change of MCBs. Such works will increase our understanding of sediment transportation and water environment management in the Yangtze River(Yang, 2014).



# 5 Summary and concluding remarks

This study presented systematic analyses of MCBs in the whole downstream of the TGD in the last three decades. It gives us the first opportunity to comprehensively understand the longitudinal and temporal dynamics of MCBs in the study area. Most of the MCBs in terms of number (98 out of 140) and total area (1172 km$^2$ out of 1291 km$^2$) were scattered in the lower reach (HK-EST) with an average interval distance of 10 km. The temporal dynamics were revealed with annual MCBs data using a statistical classification system. This classification system grouped the extracted 140 MCBs into four general types based on their area histogram distribution pattern. The four types are T1 small size (area < 2km$^2$) (50% of total number), T2 middle size (area between 2 km$^2$ and 7 km$^2$) (25%), T3 large size (area between 7 km$^2$ and 33 km$^2$) (20%), and T4 extra-large size (area > 33 km$^2$) (5%) MCBs. In each MCB type, the temporal dynamics of total number, area and shape index (i.e., LWR) were comparatively captured before and after TGD operation in order to facilitate further study of the impacts of TGD on the downstream MCBs.

MCBs number increased before TGD operation and then declined significantly after TGD operation. Among the four types of MCBs, only the T1 MCBs were experienced the number changing, and most of which happened in the lower reach. Although different types of MCBs shown overall increase trends, large MCBs tended to experience larger rate and less variation than the small MCBs. What's more, large MCBs seemed to receive less impacts of TGD on its area dynamics whereas the small MCBs likely to have gotten more influences from the TGD operation. As for the shape dynamics, MCBs with small and middle size tend to become relative shorter and wider whereas the MCBs with large and extra-large size are tend to become slimmer. Similarly, shape changes of large MCBs were more stable than those of small ones. This study implies that the scale (size) effects of MCBs on their temporal dynamics need to be paid attention in the future MCB study or MCB's practical management such as channel dredging.

The operation of TGD could have significant impact on the dynamics of MCBs. However, the study shows that their impacts could decrease with the distance to the TGD, and minimized at HK (for LWR dynamics) or JJ (area dynamics). The dynamics in the lowest JJ-EST section may have received more complicated influences such as sand mining activities and more researches on other factors are needed in the future.



**Data availability**

The raw images and gauging data are entirely publicly available data, as described in Section 2.2. The extracted MCBs data (in ESRI Shapefile format) are available by contacting the corresponding author. The finial processed data for result analysis can be found in the supplementary material.

**Author contributions**

ZW prepared the data, performed the data analysis, and wrote the manuscript. HY and CZ both contributed to the writing. CD, JC, ML, and SW contributed to the design of the study. GS and ZS offered their expertises in the image processing.

**Competing interests**

The authors declare that they have no conflict of interest.

**Acknowledgements**

This work is supported by Open Fund of State Laboratory of Information Engineering in Surveying, Mapping and Remote
Sensing, Wuhan University (No: 18R07), Open Fund of State Key Laboratory of Lake Science and Environment, Chinese Academy of Sciences (No: 2018SKL006), and by National Natural Science Foundation of China (No: 41501096, 51779241, and 41401051).

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



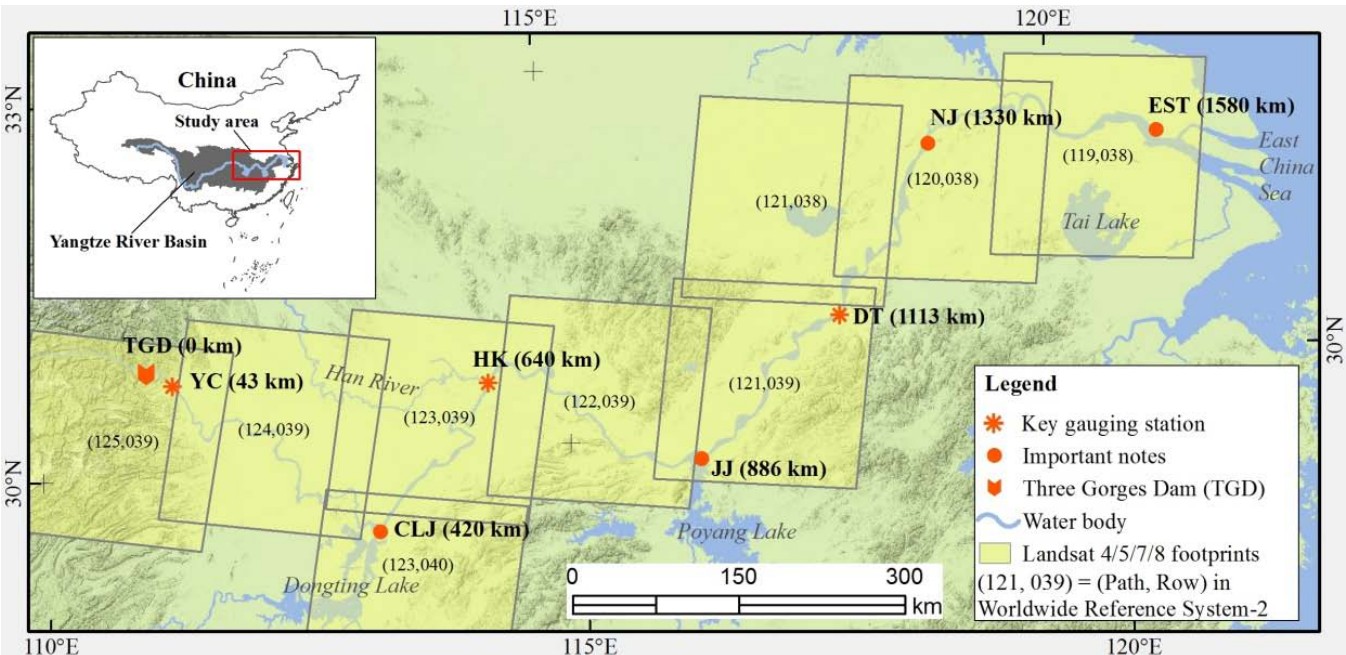

**Figure 1: Geographic location of the study area (downstream of the TGD) and its corresponding covered footprints of Landsat 4/5/7/8 images. Location names, TGD, YC, CLJ, HK, JJ, DT, NJ and EST, are acronym of Three Gorges Dam, Yichang, Chenglinji, Hankou, Jiujiang, Datong, Nanjing and estuary of the Yangtze River, respectively. The closed number following location name indicates the downward distance of that location to the TGD. For example, "YC (43km)" means Yichang gauging station is located at 43 km downstream of the TGD.**

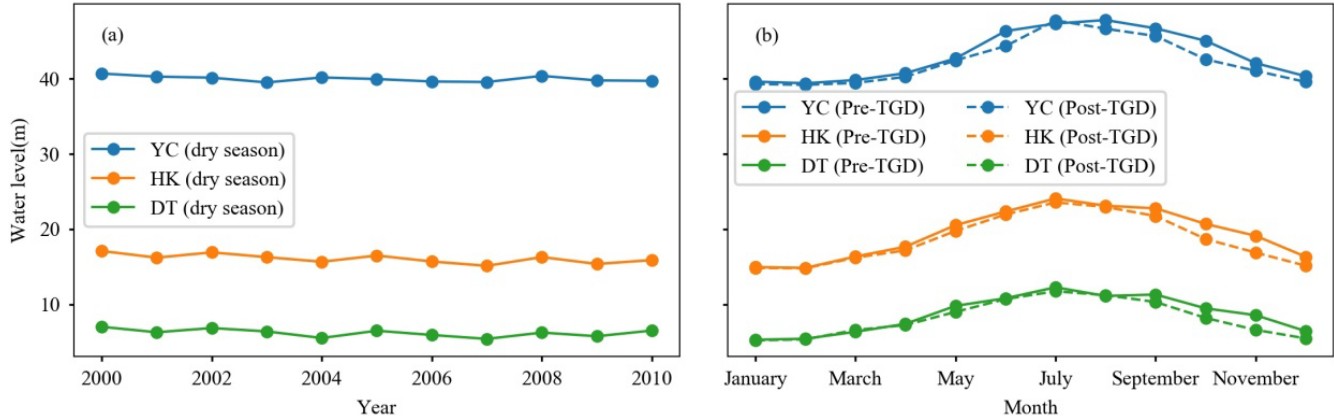

**Figure 2: Water level variations at the three gauging stations (YC, HK, and DT as shown in Fig.1) from 2000 to 2010: (a) dry season (i.e., November to March) averaged annual variations and (b) annually averaged seasonal variations for both pre-TGD (2000-2002) and post-TGD (2003-2010) periods.**

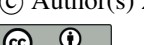



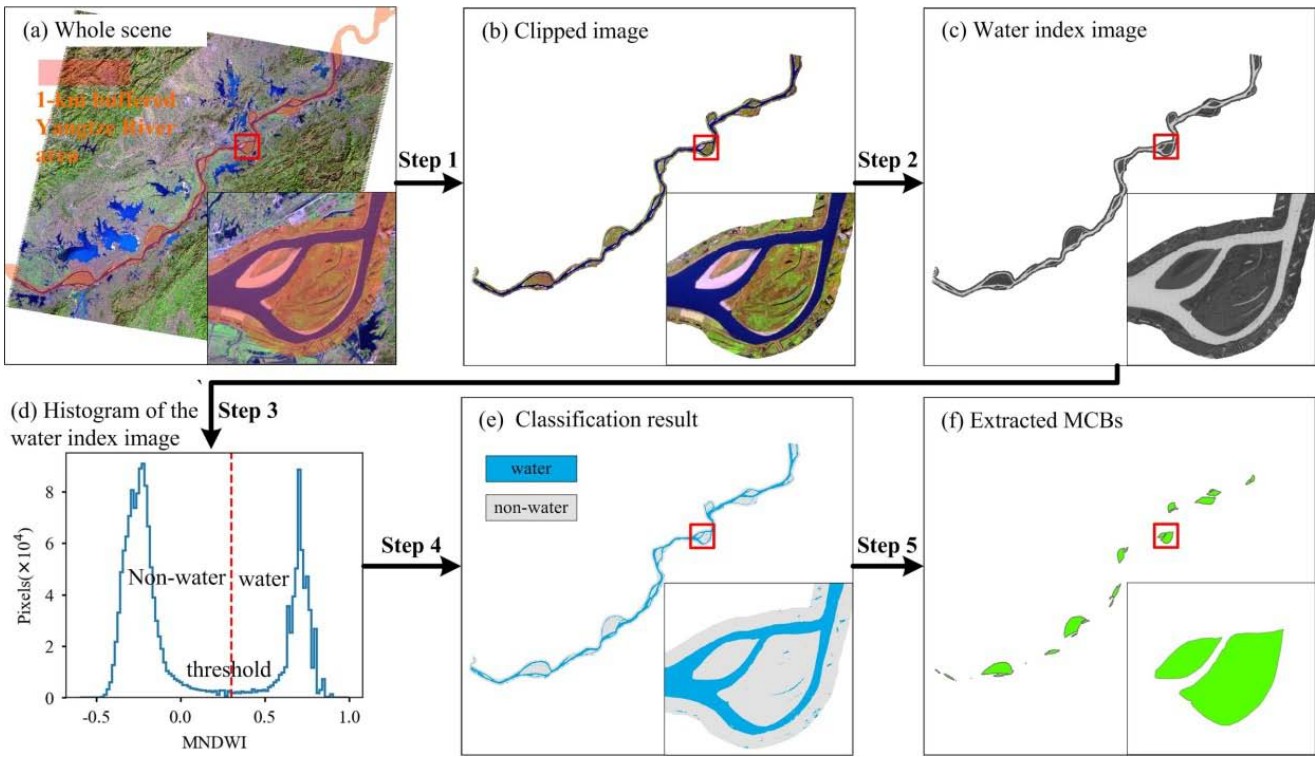

**Figure 3: Main steps for automatic extracting MCBs. (a) The whole scene of Landsat-5 image in false color (R: Band 5, G: Band 4, and G: Band 3) acquired, for example on January 14, 2010 (path=121, row=039). (b) Clipped image by the 1-km buffered Yangtze River polygon GIS data. (c) Water index image generated by the algorithm of modified normalized different water index (Xu, 2006). (d) Histogram of the water index image for determining the optimal threshold. (e) Raw classification result by applying threshold method. (f) Extracted MCBs from the raw classification result. Steps 1-5 are detailed below.**

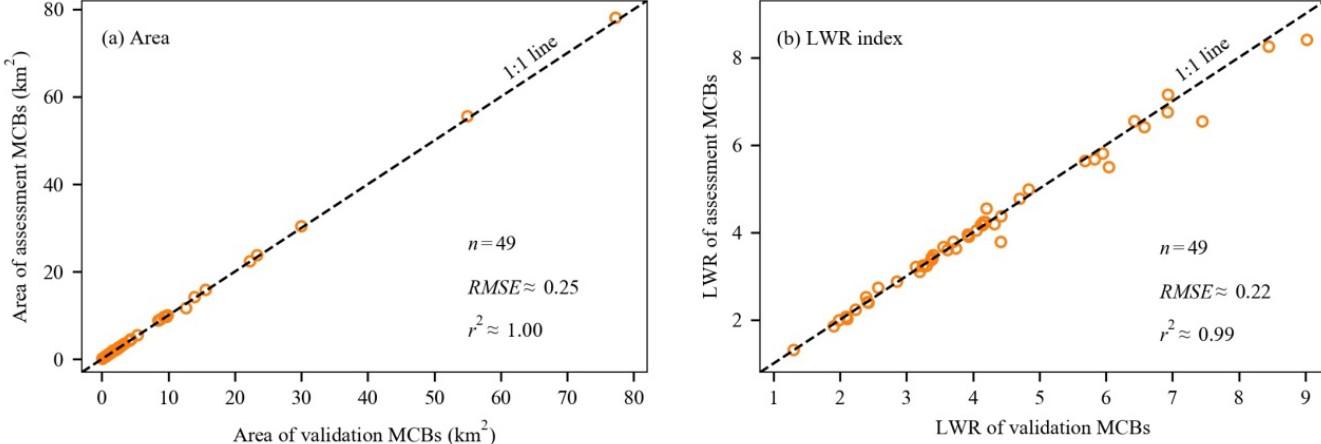

**Figure 4: Accuracy assessment of MCBs from perspectives of area (a) and LWR index (b).**



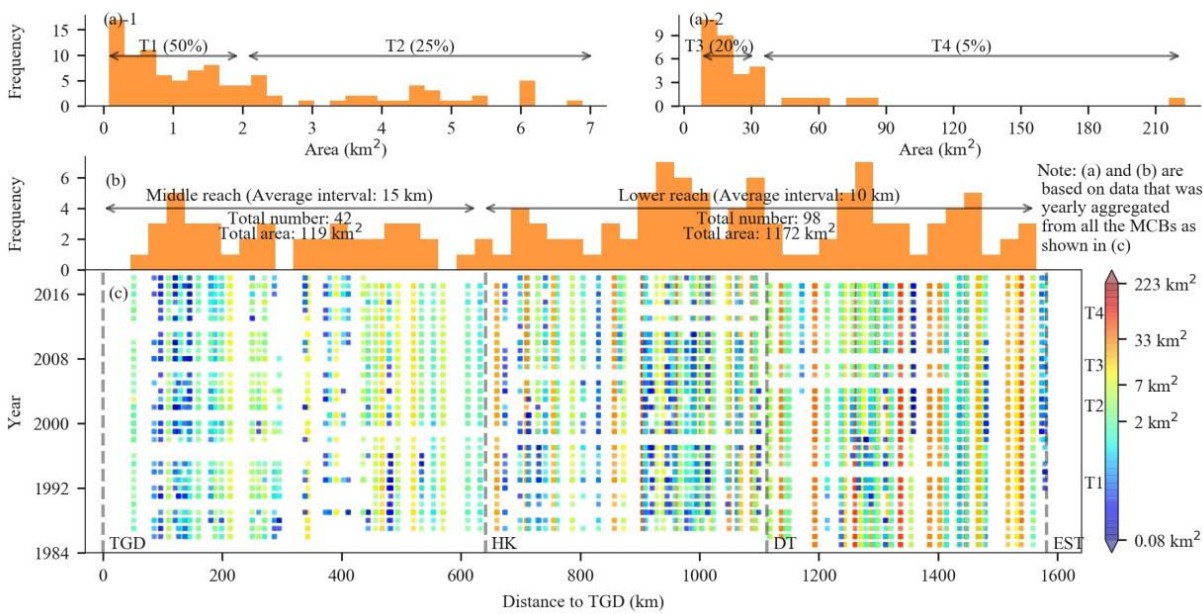

**Figure 5: Overview of MCBs that extracted from Landsat images. Figs. (a)-1 and (a)-2 are area histograms of MCBs with consecutive area arranges. (b) Distance-to-TGD histogram of MCBs, and (c) Longitudinal and temporal distribution of all MCBs with different areas as indicated by their colors. T1, T2, T3, and T4 stand for four different MCBs types as explained in Table 2.**

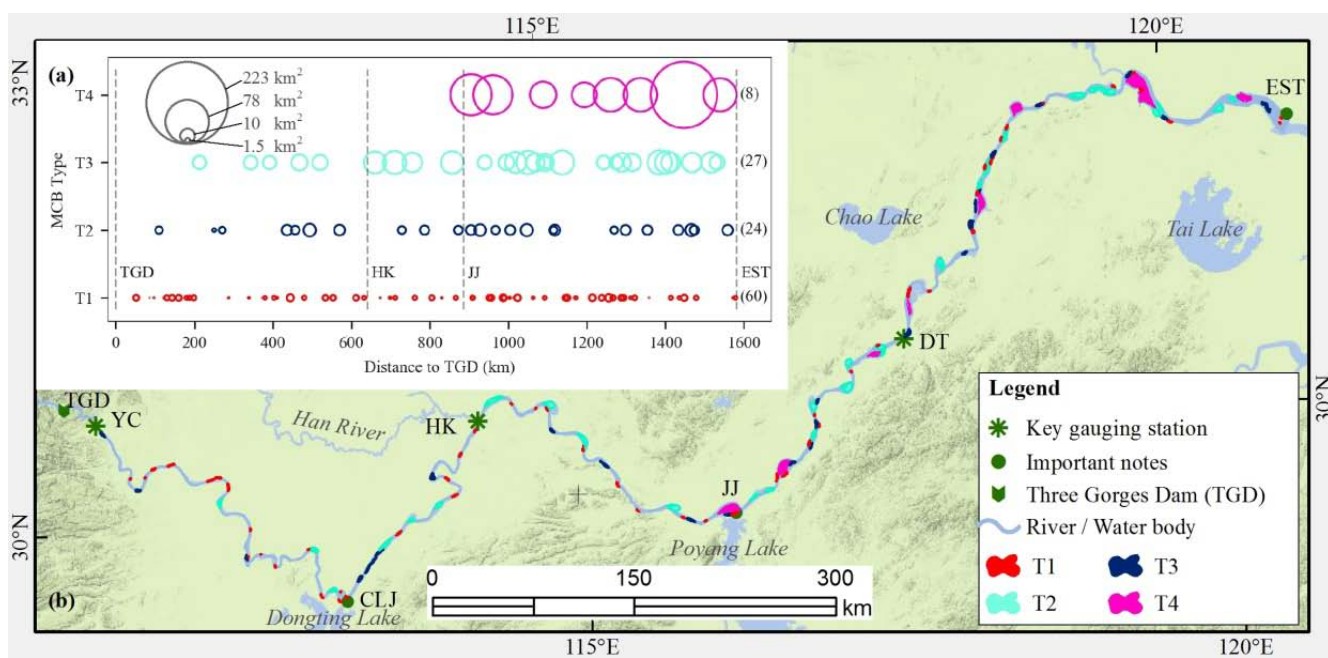

**Figure 6: Longitudinal distribution of MCBs along downstream of the TGD in 2016.**





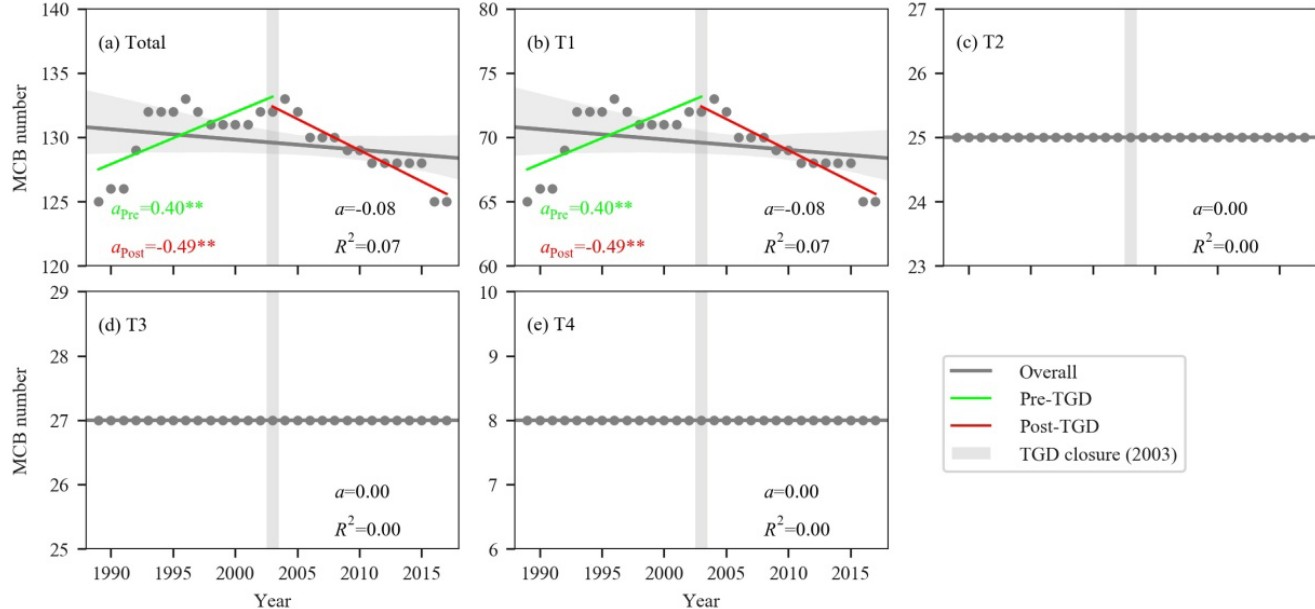

**Figure 7: Temporal dynamics of MCB number for different groups of MCBs: (a) Total MCBs, (b) T1 MCBs, (c) T2 MCBs, (d) T3 MCBs, and (e) T4 MCBs. The regression parameters *a* and $R^2$ stand for regression slope and coefficient of determination, respectively. Slope value with "*" and "**" indicated the corresponding *p*-value < 0.05 and < 0.01, respectively.**

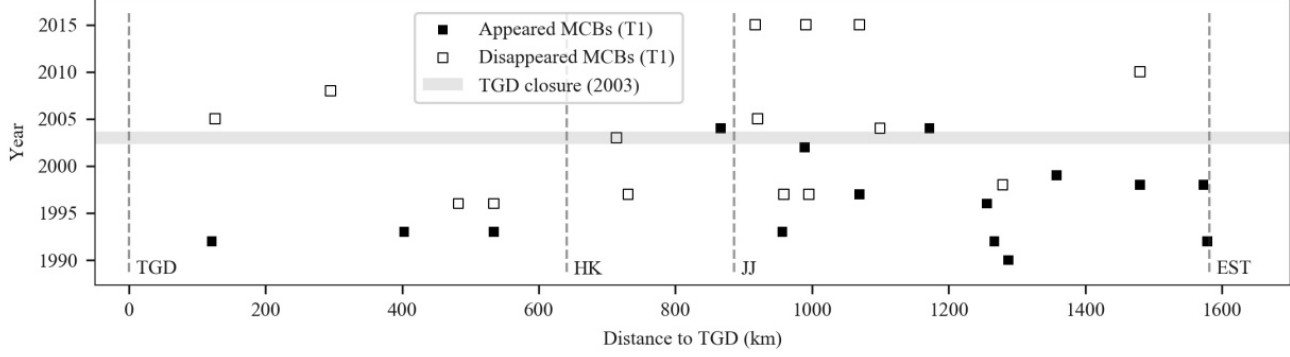

10 **Figure 8: Longitudinal -temporal distribution of number change events**



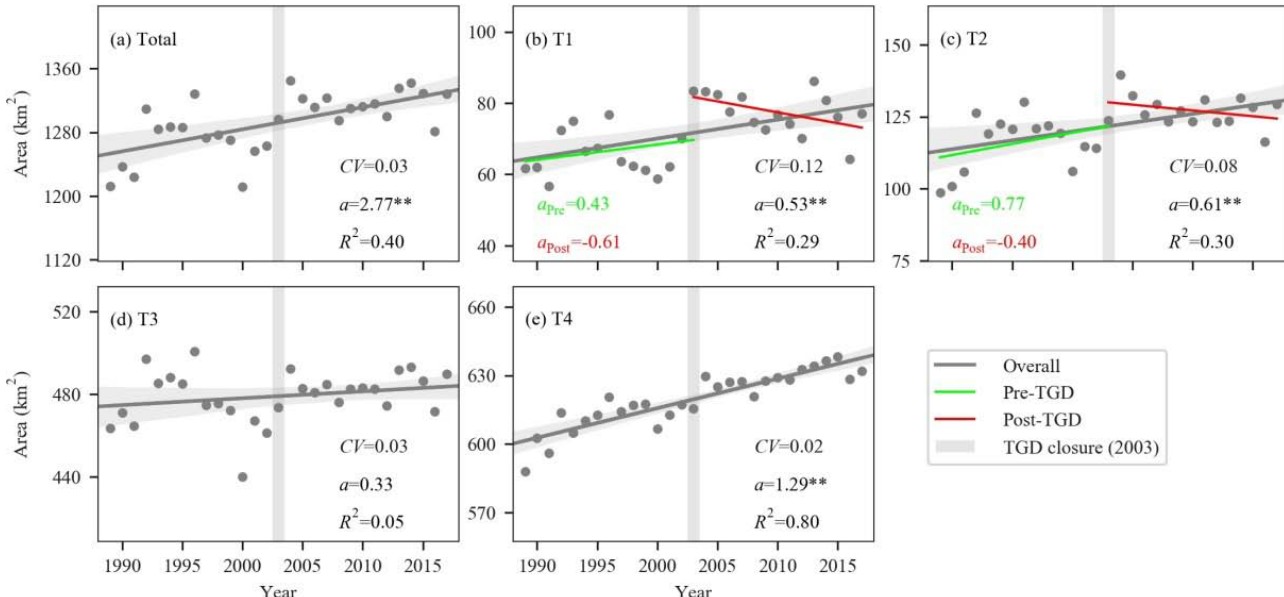

**Figure 9: Area temporal dynamics of different groups of MCBs: (a) Total MCBs, (b) T1 MCBs, (c) T2 MCBs, (d) T3 MCBs, and (e) T4 MCBs. The regression parameters $a$ and $R^2$ stand for regression slope and coefficient of determination. Slope values with "*" and "**" indicate the corresponding $p$-value $< 0.05$ and $< 0.01$, respectively. $CV$ strands for coefficient of variation which is defined as the ratio of the standard deviation to the mean and here it indicates the extent of variability in relation to the mean area of the MCBs.**

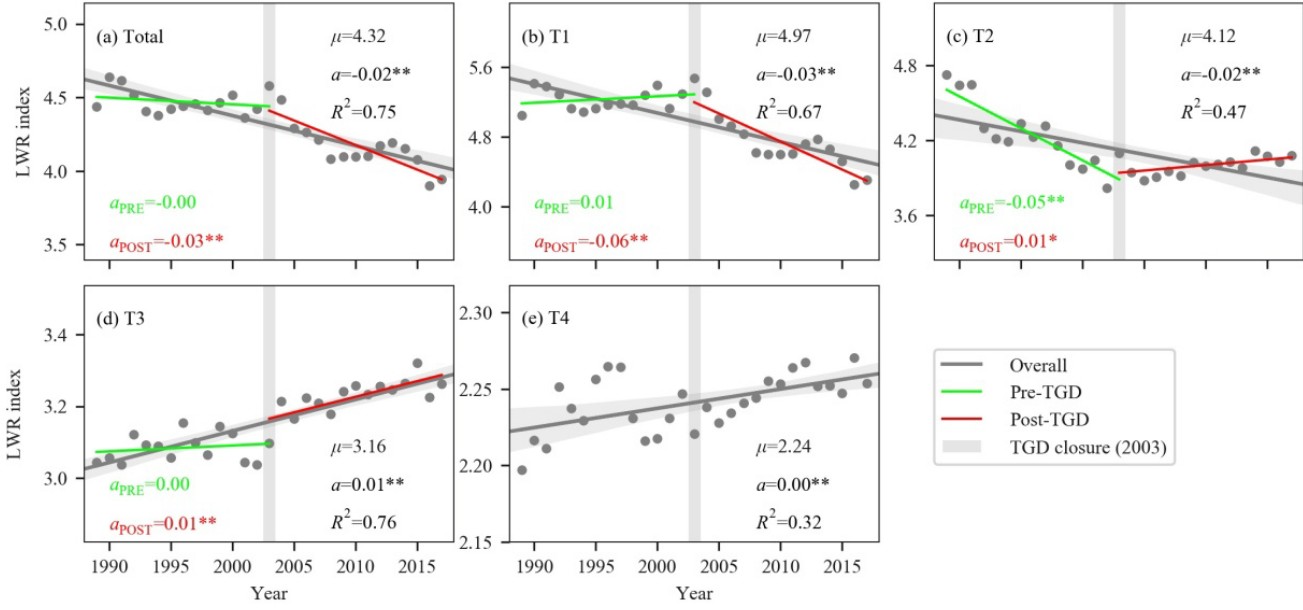

**Figure 10: LWR temporal dynamics of different groups of MCBs: (a) Total MCBs, (b) T1 MCBs, (c) T2 MCBs, (d) T3 MCBs, and (e) T4 MCBs. The regression parameters $a$ and $R^2$ stand for regression slope and coefficient of determination. Slope values with "*" and "**" indicate the corresponding $p$-value $< 0.05$ and $< 0.01$, respectively. Parameter $\mu$ stands for mean value of LWR.**



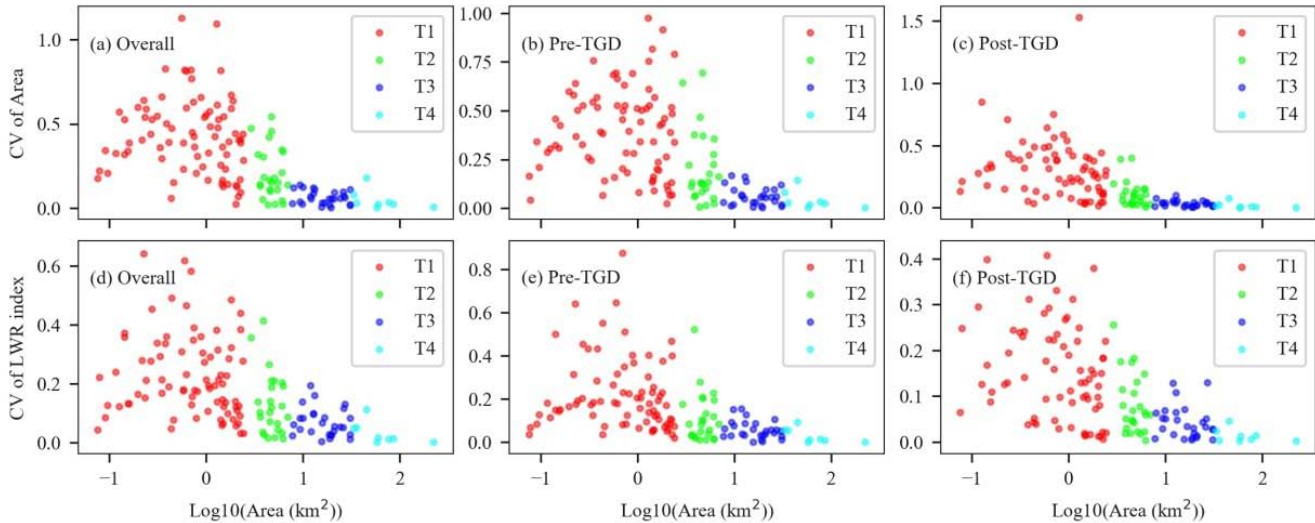

**Figure 11: CV-indicated temporal stability of all individual MCBs changed with their sizes (indicated by logarithmized area) in different periods (overall, pre-TGD, and post-TGD).**

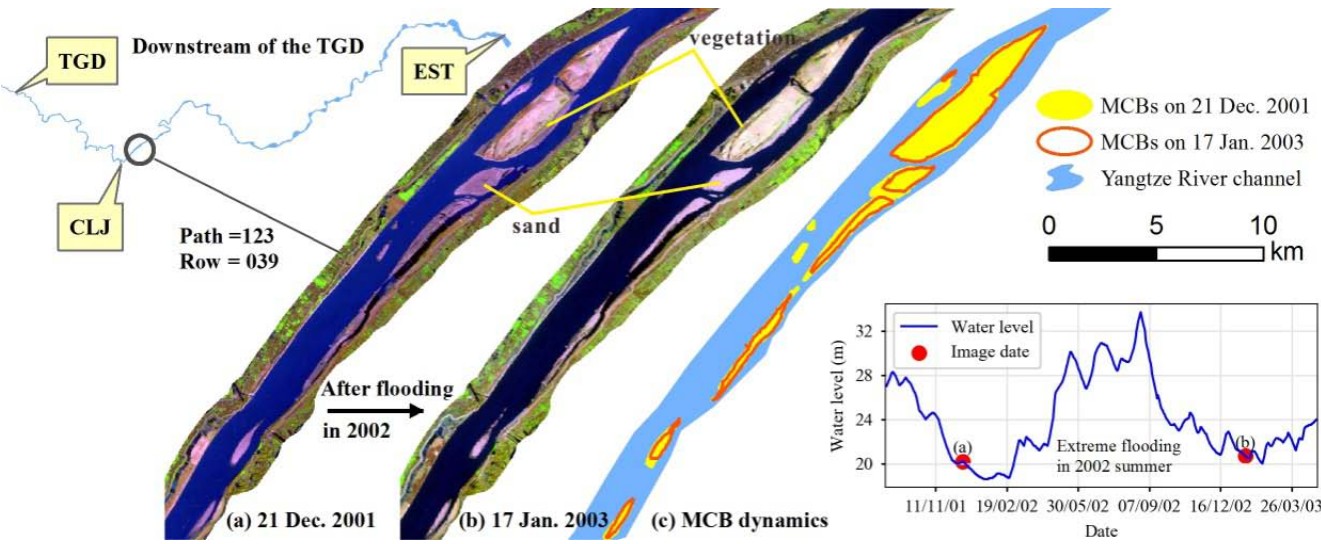

**Figure 12:  An example of performances of MCBs with different sizes in both area and shape changes after a flooding event.**



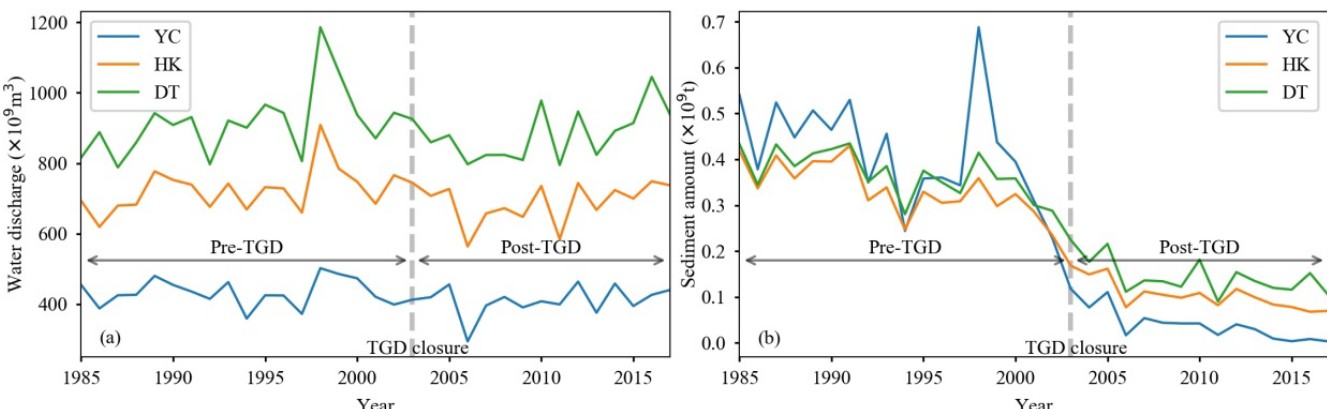

**Figure 13: Flow and sediment regimes in the pre-TGD and post-TGD periods in the three key gauging stations.**

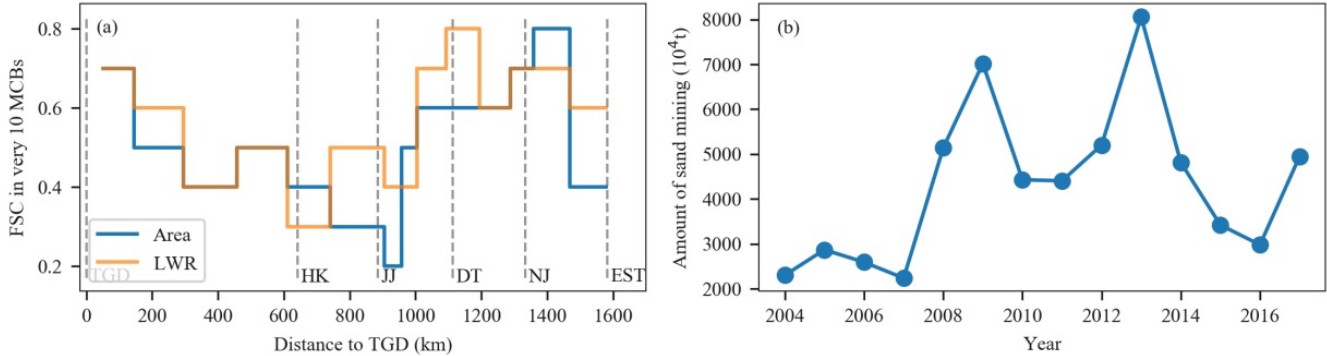

**Figure 14: Frequency of structural changes (FSC) with distance to the TGD (a). Total amount of sand mining in the downstream of the TGD (b).**

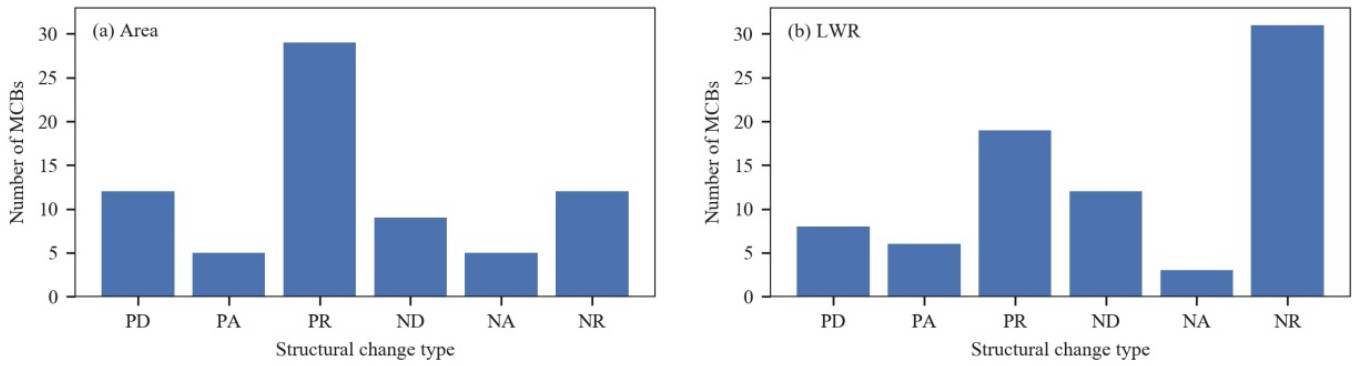

**Figure 15: Different structural change types (according to Supplementary Table S2) and their corresponding MCBs numbers for temporal dynamics of area (a) and LWR (b). For the terms, PD, PA, PV, ND, NA, and NV, the first letter "P" or "N" stands for positive trend or negative trend in the pre-TGD period; the second letter "D", "A" or "R" in each term indicates in a decelerate trend, accelerate, or a reverse trend in the post-TGD period. For example, "PR" in subfigure (a) grouped the MCBs which have structural change in the area temporal dynamics with positive trend pre-TGD and negative trend post-TGD.**





**Table 1. HSR images and spatially corresponding Landsat images for accuracy assessment**

| Google Earth based HSR images | | Landsat images | | | |
| --- | --- | --- | --- | --- | --- |
| Source | Acquisition date | Sensor | Path | Row | Acquisition date |
| DigitalGlobe | 2010-12-02 | Landsat 5 | 119 | 038 | 2010-12-01 |
| CNES Airbus | 2013-12-10 | Landsat 8 | 119 | 038 | 2013-12-11 |
| DigitalGlobe | 2003-01-04 | Landsat 5 | 120 | 038 | 2003-01-07 |
| CNES Airbus | 2013-11-15 | Landsat 8 | 120 | 038 | 2013-11-20 |
| DigitalGlobe | 2014-11-18 | Landsat 8 | 120 | 038 | 2014-11-16 |
| CNES Airbus | 2017-12-12 | Landsat 8 | 120 | 038 | 2017-12-11 |
| DigitalGlobe | 2008-12-10 | Landsat 5 | 121 | 039 | 2008-12-08 |
| DigitalGlobe | 2015-02-13 | Landsat 8 | 121 | 039 | 2015-02-08 |
| DigitalGlobe | 2017-12-19 | Landsat 8 | 121 | 039 | 2017-12-21 |
| DigitalGlobe | 2017-12-26 | Landsat 8 | 122 | 039 | 2017-12-21 |
| CNES Airbus | 2018-01-11 | Landsat 8 | 122 | 039 | 2018-01-11 |
| DigitalGlobe | 2004-02-13 | Landsat 5 | 123 | 039 | 2004-02-14 |
| DigitalGlobe | 2006-12-19 | Landsat 5 | 123 | 039 | 2006-12-17 |
| DigitalGlobe | 2017-02-16 | Landsat 8 | 123 | 039 | 2017-02-13 |
| CNES Airbus | 2015-01-01 | Landsat 8 | 124 | 039 | 2014-12-31 |
| DigitalGlobe | 2016-12-05 | Landsat 8 | 124 | 039 | 2016-12-08 |
| CNES Airbus | 2017-01-22 | Landsat 8 | 124 | 039 | 2017-01-16 |
| DigitalGlobe | 2017-12-24 | Landsat 8 | 124 | 039 | 2017-12-17 |
| DigitalGlobe | 2018-01-09 | Landsat 8 | 124 | 039 | 2018-01-09 |

**Table 2. Four different types of MCBs.**

| Type | Explanation | Area range ($km^2$) | | Proportion |
| --- | --- | --- | --- | --- |
| | | Minimum | Maximum | |
| T1 | Small size | 0.02[a] | 2 | 50% |
| T2 | Middle size | 2 | 7 | 25% |
| T3 | Large size | 7 | 33 | 20% |
| T4 | Extra-large size | 33 | 223[b] | 5% |

Note: [a] Only MCB with area larger than 0.02 $km^2$ was considered as "true" MCBs as described in the Step 5 in extracting

5   MCBs. [b] 223 $km^2$ was the largest area among the MCBs' area data.