# Peer review of "Three-decadal dynamics of mid-channel bars in downstream of the Three Gorges Dam, China"

_Hydrology and Earth System Sciences, 2019_

## Referee Comment (RC1) · Anonymous Referee #1 · 25 Jun 2019

This study focuses on the morphological dynamics of mid-channel bars (MCBs) downstream the Three Gorges Dam (TGD) in China. The authors extracted the size and shape of MCBs from three decades of Landsat images and identified them to follow their temporal dynamics. MCBs are classified depending on their size, from small (< 2 km2) to extra-large (> 33 km2). While small MCBs are equally distributed all along the Yangze river downstream the TGD, large and extra-large MCBs are mainly located in the lower reach (downstream the Jiujiang gauge station). The authors then evaluated the impact of the TGD closure in 2003 on the temporal dynamics of MCBs. Main results are: 1) a decreasing number of small MCBs after the TGD closure, mostly in the lower reach, 2) an increasing trend of all MCB classes, slightly impacted by the TGD closure, 3) an opposite change in shape for small and large MCBs, the latter being more stable

and less impacted by the TGD closure. Overall, the study is well conducted and the paper well organized. Although English is not my first language, I would suggest to the authors to have the manuscript reread and corrected by an English native speaker. Apart from that, I found the paper very interesting and well suited to HESS, and I only have a few minor remarks.

P5L25. Banks of MCBs may have quite a small slope and MCBs area could then be very sensitive to water level. Do the authors have characterized this sensitivity?

P7L14. Is there a quantification of the bimodal characteristic of each image? In other words, did the MHBM work well for all images?

P8L16. Is the length L of MCBs defined as the length in the flow direction or is it simply the longest dimension?

P9L12-13. Change Fig 3 to Fig 4.

P9L20-22. Do percentages correspond to the total number or total area? It seems from the conclusion (P15L7) it is percentages of the total number.

P10L4. The authors mention the density of MCBs in the text, whereas they use the term "frequency" in Fig. 5. Please be consistent.

P10L6-8. The average interval is 15 km for the middle reach and 10 km for the lower reach. I understand that the density of MCBs is higher in the lower reach, but for consistency I would have kept the same averaging length.

P10L7. "upper reach" should be "middle reach", given the definition provided in P4L18.

P10L7. The authors stated that there are twice more MCBs in the lower reach than in the middle reach. But given than the lower reach is twice longer it is not clear to me if the density is that higher in the lower reach. The authors should choose a proper metrics to conclude on this.

P11L22. Please define mu.

P11L23. "with the increasing MCBs size" seems confusing (section 3.4.2 and Fig 9a show increasing MCBs size, but in terms of positive area trends). Please rephrase.

P12L20. Which type of vegetation do the authors refer to? Is it mostly grass with shallow roots or trees with deeper roots?

P13L3. Change Fig 12 to Fig 13. Same P13L6.

P13L4-5. Could the authors add a figure showing seasonal averages over pre- and post-TGD closure periods?

P13L10-15. An expected impact of TGD operations is the flood peak reduction. Could the authors show a graph depicting the evolution of extreme discharges before and after the TGD closure?

P13L21. Could the authors expand a little bit on the analysis by Wang et al. (2013) and Yuan et al. (2012)?

P13L29-30. Fig 14b shows sand mining activities downstream of TGD, it is not shown that "those sand mining activities mainly focused on the lower reach".

P14L11. Could the authors provide a quantification of the error due to water level variations, or at least give an order of magnitude?

P14L20-22. What is the uncertainty related to Landsat spatial resolution? Could this be approximated by computing the impact on the area of adding/removing 1 pixel on the MCB boundary?

Fig 3. Please remove the last sentence of the figure caption ("Steps 1-5 are detailed below").

Fig 5. Please define frequency (or density to be consistent with the text) in this context. Also, does each dot in Fig 5c represent a single MCB? If yes, this should be said.

Fig 11. I would suggest to keep same y-axis bounds on each graph for a better comparison (at least for each line).

[Figure]

---

## Referee Comment (RC2) · Anonymous Referee #2 · 14 Aug 2019

The present manuscript introduces a description of the evolution of mid-channel bars (MCB) downstream of the Three Gourges Dam (TGD) in China. Authors used Landsat archive images in order to identify the MCB and describe their evolution in time. The study highlighted the impact of the TGD on sediment transport and also demonstrated that the small bars are more vulnerable and have the highest variability. The manuscript is quite easy to ready and well structured, but it is mainly a qualitative study based on observations. I did not find a significant effort in the description of the dynamics driving the shift of the evolution of the bars. Given the macroscopic change induced by the construction of dam it would be interesting to give additional insights on the evolution of this system trying to understand how and when other bars will be affected. In this a numerical analysis on the evolution of the river system may be extremely useful.

In my personal perspective, the manuscript does not contain any relevant conclusion. The result are quite obvious, the methodology is not innovative and the conclusion irrelevant. It is not surprising to observe that small fluvial bars are the most likely to disappear after a flood. Therefore, I do not understand what is the innovative contribution of the present manuscript beside the construction of the dataset.

The main contribution that may have some value is represented by the dataset reconstructed about the morphological evolution of the river system. Such a data base may be relevant and useful, but I believe that the author should consider to change journal and eventually propose the manuscript for Earth System Science Data or Data in Brief. I do not think that the manuscript can be accepted in HESS with any attempt to provide a physical explanation of the dynamics of fluvial bar and of their mutual interaction with the dam management.

---

## Author Comment (AC1) · 1 Sep 2019

Dear Editor,

We would like to thank you and the reviewer. The reviewer has raised a number of important comments. We have revised our manuscript to address the comments. Below are our point-by-point responses to the reviewer's comments.

**Response to Reviewer #1**

(1)  This study focuses on the morphological dynamics of mid-channel bars (MCBs) downstream the Three Gorges Dam (TGD) in China. The authors extracted the size and shape of MCBs from three decades of Landsat images and identified them to follow their temporal dynamics. MCBs are classified depending on their size, from small (< 2 km2) to extra-large (> 33 km2). While small MCBs are equally distributed all along the Yangze river downstream the TGD, large and extra-large MCBs are mainly located in the lower reach (downstream the Jiujiang gauge station). The authors then evaluated the impact of the TGD closure in 2003 on the temporal dynamics of MCBs. Main results are: 1) a decreasing number of small MCBs after the TGD closure, mostly in the lower reach, 2) an increasing trend of all MCB classes, slightly impacted by the TGD closure, 3) an opposite change in shape for small and large MCBs, the latter being more stable and less impacted by the TGD closure. Overall, the study is well conducted and the paper well organized. Although English is not my first language, I would suggest to the authors to have the manuscript reread and corrected by an English native speaker. Apart from that, I found the paper very interesting and well suited to HESS, and I only have a few minor remarks.

**Response:** We thank the reviewer for making great efforts in reviewing the manuscript and positive comments. We made careful revision accordingly. Please find detailed responses to the comments below. In addition, we have checked English throughout the manuscript and had it polished by a native English speaker. We sincerely hope the revised version will meet the publication standard of HESS.

(2)  P5L25. Banks of MCBs may have quite a small slope and MCBs area could then be very sensitive to water level. Do the authors have characterized this sensitivity?

**Response:** Thanks for the question. According to our *in situ* observation, banks around the tail parts of MCBs (downstream parts) often present gentle slopes, while banks around the head parts of MCBs (upstream parts) are often in steep slopes due to the general deposition (at tail) and erosion (at head) pattern (Fig. R1). We agree that the MCBs areas are sensitive to water levels. That is why we have carefully considered this sensitivity by minimizing water level variations through selecting images only acquired in dry season as illustrated in Fig. 2. Following the suggestion, we have analyzed the sensitivity of the water level change on the MCBs areas and inserted the result into Fig. 2 (Fig. 2c). It is spotted that 0.5 m change of water level could cause about 0.1 km$^2$ change of MCBs' area.

[Figure]

**Figure R1: A typical MCB (drone image overlapped by shaded relief) in Yangtze River.**

[Figure]

**Fig. 2: Water level variations at the three gauging stations (YC, HK, and DT as shown in Fig.1) from 2000 to 2010: (a) dry season (i.e., November to March) averaged annual variations; (b) annually-averaged seasonal variations for both pre-TGD (2000-2002) and post-TGD (2003-2010) periods; and (c) Sensitivity of MCB area variation to the water level variation.**

(3) P7L14. Is there a quantification of the bimodal characteristic of each image? In other words, did the MHBM work well for all images?

**Response:** Thanks for the inspiring point. To our best knowledge, we could not find an effective and automatic way to check the bimodal characteristic in a histogram. But we visually checked the histogram of each water index image and found that 498 out 553 images had bimodal histogram patterns (Supplementary folder "Histogram of MNDWI images"). For the other 5 images with non-bimodal histograms, the MCBs were manually extracted and related revision was made in the main text Page 7 Line 10-12.

"*There are 498 out of 553 MNDWI images illustrating distinct bimodal histogram patterns which were very suitable for the application of MNBM (see Supplementary folder "Histogram of MNDWI images"). For the other 5 MNDWI images with non-bimodal histograms, the MCBs were extracted manually*"

Moreover, the effects of MHBM method on extracting MCBs were analyzed in Section 3.1, which shows that the MHBM worked well for the involved MNDWI images.

(4) P8L16. Is the length L of MCBs defined as the length in the flow direction or is it simply the longest dimension?

**Response:** Sorry for the lack of clarity at this point. The $L$ was calculated from the length dimension of the convex rectangle of MCB. The direction of such length dimension is generally similar to the flow direction. The corresponding text was added in the main text Page 8 Line14.

"*where L and W represent the length and width of the convex rectangle of an MCB, respectively.*"

(5) P9L12-13. Change Fig 3 to Fig 4.

**Response:** Sorry for the error. A correction was made.

(6) P9L20-22. Do percentages correspond to the total number or total area? It seems from the conclusion (P15L7) it is the percentages of the total number.

**Response:** Sorry for the confusion. It refers to the percentage of total MCBs number. The revision was made accordingly in the main text Page 9 Lines 15-17.

"*Calculated by MCB numbers, 50% of the MCBs are smaller than 2 km$^2$; 25% of MCBs are with area range 2 - 7 km$^2$; 20% number of MCBs are with area range 7 - 33 km$^2$; and only 5% of MCBs are with area larger than 33 km$^2$.*"

(7) P10L4. The authors mention the density of MCBs in the text, whereas they use the term "frequency" in Fig. 5. Please be consistent.

**Response:** The term "Frequency" was changed to "Number" in Fig. 5.

[Figure]

**Figure 5:** **Overview of MCBs that extracted from Landsat images. Figs. (a)-1 and (a)-2 are area histograms of MCBs with consecutive area arranges. (b) Distance-to-TGD histogram of MCBs, and (c) Longitudinal and temporal distribution of all MCBs (each dot represents for a single MCB) with different sizes as indicated by their colors. T1, T2, T3, and T4 stand for four different MCBs types as explained in Table 2.**

(8)  P10L6-8. The average interval is 15 km for the middle reach and 10 km for the lower reach. I understand that the density of MCBs is higher in the lower reach, but for consistency I would have kept the same averaging length.

**Response:** Thanks for the suggestion. Here, the average interval is served a quantitative indicator of MCBs densities and it is calculated as *(Length of reach) / (total MCBs number in that reach)*. So the smaller average interval between MCBs, the higher distribution density of MCBs. The paragraph has been rewritten. It is much clearer in this version.

"*In the middle reach, there are 42 MCBs (30% of total MCBs number) scattered along the channel with an average interval of 15 km from 1985 to 2019. The sum of the annually-averaged MCBs area in the middle reach was 119 km², accounted for 9% of the total MCBs area in the entire downstream (Fig. 5b). By contrast, there were 98 MCBs (70% of total MCBs number) along the lower reach and distributed within an average interval of 10 km, relatively closer (or higher in density) than that in the middle reach. The sum of the annually-averaged MCBs area in lower reach was 1172 km², accounted for 91% of the total MCBs area. These distinct longitudinal distribution patterns highlight the importance of lower reach in the analysis of MCBs from both quality and quantity perspectives.*"

(9)  P10L7. "upper reach" should be "middle reach", given the definition provided in P4L18.

**Response:** Sorry for the error. Corrected as suggested.

(10)  P10L7. The authors stated that there are twice more MCBs in the lower reach than in the middle reach. But given than the lower reach is twice longer it is not clear to me if the density is that higher in the lower reach. The authors should choose a proper metrics to conclude on this.

**Response:** Sorry for the lack of clarity at this point. In this study, the density is calculated and presented by the average interval. Smaller average interval indicates higher density appearance of MCBs. Related to comment 8, the paragraph has been rewritten (Please also see our response to comment 8). We hope it is much clearer in this version.

(11)  P11L22. Please define mu.

**Response:** Sorry for the lack of clarity at this point. The symbol $\mu$ in Fig. 10 represents the mean value of the LWR index. Added as suggested.

(12)  P11L23. "with the increasing MCBs size" seems confusing (section 3.4.2 and Fig 9a show increasing MCBs size, but in terms of positive area trends). Please rephrase.

**Response:** Sorry for the confusion. The sentence has been rewritten.
     "In *general, the larger MCBs sizes are, the smaller LWR values are (i.e., μ values in Fig. 10), namely, 4.97 (T1) > 4.12 (T2) > 3.16 (T3) > 2.24(T4)."*

(13)  P12L20. Which type of vegetation do the authors refer to? Is it mostly grass with shallow roots or trees with deeper roots?

**Response:** Thanks for the question. According to our *in situ* observation, it's unlikely that grass with shallow roots can survive on sand bars under frequent floodings. Although we cannot recognize the exact vegetation species on the bar (Fig. 12) in 2002, the vegetation would most likely be shrub or grass with deep roots. After 15 years development, the bar has been entirely covered by vegetation (probably the *Phragmites australis*) according to the recent (i.e., May 27th 2019) high spatial resolution satellite image on Google Earth.

(14)  P13L3. Change Fig 12 to Fig 13. Same P13L6.

**Response:** Sorry for the errors. Corrected as suggested.

(15)  P13L4-5. Could the authors add a figure showing seasonal averages over pre- and post-TGD closure periods?

**Response:** Thanks for the good suggestion. The figure showing seasonal averages pre- and post-TGD was added (Fig. 13c).

[Figure]

**Figure 13:** Flow and sediment regimes in the pre-TGD and post-TGD periods in the three key gauging stations. (a) Annual water discharge from 1985 to 2017, (b) Annual sediment amount from 1985 to 2017, and (c) Monthly average water discharge from 1989 to 2002 (Pre-TGD) and from 2003 to 2017 (Post-TGD).

(16) P13L10-15. An expected impact of TGD operations is the flood peak reduction. Could the authors show a graph depicting the evolution of extreme discharges before and after the TGD closure?

**Response:** Thanks for the suggestion. The new Fig. 13c (please see the response to the last comment) was added to show the reduction of peak water discharge in flood season after the TGD closure.

(17) P13L21. Could the authors expand a little bit on the analysis by Wang et al. (2013) and Yuan et al. (2012)?

**Response:** Thanks for the suggestion. More details were added in the main text to elaborate on these two references.

"*For example, Wang et al., (2013) found that the operation of the TGD reduced the water level in the whole downstream, but the effects were mainly laid on the immediate downstream reach and generally diminished in a longitudinal direction from the TGD to the estuary. Regarding the channel dynamics, Yuan et al., (2012) evidenced that the channel scour appeared in the post-dam period and the largest strength happened at immediate downstream and then decreased longitudinally.*"

(18) P13L29-30. Fig 14b shows sand mining activities downstream of TGD, it is not shown that "those sand mining activities mainly focused on the lower reach".

**Response:** Sorry for the lack of clarity at this point. The data in Fig. 14b were extracted from reports of Changjiang Water Resources Commission (2003-2017). The data were the sum of sand mining in the entire downstream of the TGD, and the reports mentioned that most of the sand mining activities happened in the lower reach. Therefore, we cited the text "those sand mining activities mainly focused on the lower reach" from the report. To avoid confusion, we have made a revision in the main text.

*"Those sand mining activities were reported mainly happened on the lower reach, particularly in the JJ-EST reach (Changjiang Water Resources Commission, 2003-2017)."*

(19) P14L11. Could the authors provide a quantification of the error due to water level variations, or at least give an order of magnitude?

**Response:** thanks for the suggestion. Related to Comment#2, the information was added as shown in Fig. 2c (see our response to the comment#2).

(20) P14L20-22. What is the uncertainty related to Landsat spatial resolution? Could this be approximated by computing the impact on the area of adding/removing 1 pixel on the MCB boundary?

**Response:** Thanks for the question. In this study, two types of uncertainties are related to the relative coarse Landsat spatial resolution (30 m). The first one is the inherent mixed pixel challenge in remote sensing applications. This type of uncertainty is what the reviewer concerned about. The boundary of MCBs often consists of mixed water and non-water components. In the classification process, the mixed pixels were partially classified as non-water area and some were classified as water areas. Therefore, it could be one source of error for the extracted MCBs. However, the error of MCBs has been assessed in Section 3.1. To our best knowledge, the uncertainty cannot be simply approximated by adding/removing 1 pixel on the MCB boundary.

The second type of uncertainty brought by the relative coarse Landsat spatial resolution (30 m) is that the automatically extracted patches with area less than 0.02 $km^2$ were considered as man-made features (e.g., ships) and not included in the analysis. This simple rule of distinguishing MCBs from man-made features is not always suitable and some uncertainties could be introduced as a result. The reason for making such a simple rule is that the small patches with area less than 0.02 $km^2$ are less than 22 pixels (900 $m^2$ for one pixel) which were hard to form a clear shape that could help authors to distinguish MCBs from man-made features. In future studies with available high spatial resolution images, MCBs with area less than 0.02 $km^2$ could be identified visually.

(21) Fig 3. Please remove the last sentence of the figure caption ("Steps 1-5 are detailed below").

**Response:** It was deleted as suggested.

(22) Fig 5. Please define frequency (or density to be consistent with the text) in this context. Also, does each dot in Fig 5c represent a single MCB? If yes, this should be said.

**Response:** Fig. 5 has been modified as suggested. Each dot in Fig 5c represents a single MCB and the caption has been corrected.

(23) Fig 11. I would suggest to keep same y-axis bounds on each graph for a better comparison (at least for each line).

**Response:** Revised as suggested.

[Figure]

**Figure 11: CV-indicated temporal stability of all individual MCBs changed with their sizes (indicated by logarithmized area) in different periods (overall, pre-TGD, and post-TGD).**

---

## Author Comment (AC2) · 1 Sep 2019

Dear Editor,

We thank you and the reviewer. The reviewer has raised some important concerns. We have tried our best to answer his/her concerns. Below are our point-by-point responses to the reviewer's comments.

**Response to Reviewer #2**

(1)   The present manuscript introduces a description of the evolution of mid-channel bars (MCB) downstream of the Three Gorges Dam (TGD) in China. Authors used Landsat archive images in order to identify the MCB and describe their evolution in time. The study highlighted the impact of the TGD on sediment transport and also demonstrated that the small bars are more vulnerable and have the highest variability. The manuscript is quite easy to read and well structured, but it is mainly a qualitative study based on observations. I did not find a significant effort in the description of the dynamics driving the shift of the evolution of the bars. Given the macroscopic change induced by the construction of dam it would be interesting to give additional insights on the evolution of this system trying to understand how and when other bars will be affected. In this, a numerical analysis on the evolution of the river system may be extremely useful.

**Response:** We appreciate the comments and concerns, which indicate some unclarity in the manuscript that needs to be improved. Assessing whether the manuscript is a qualitative or quantitative study is relatively unimportant but the main findings of this study are based on straightforward but quantitative analysis. We elaborate on it as follows:

As in the Method section, we explained how the analytical data (i.e., the area and shape index of MCBs) prepared, how the analysis methods (regression and structural breaks test) were conducted.

As in the Result section, we firstly quantitatively assessed the accuracy of the area and shape index of the MCBs data (see Section 3.1). Secondly, we quantified the basic statistical parameters of the MCBs and classified MCBs into four categories based on their statistical area histogram patterns (see Section 3.2). Thirdly, we quantitatively stated the spatial variation (Longitudinal distribution) of the MCBs in Section 3.3. In section 3.4, we quantitatively analyzed the temporal variations of the MCBs in terms of number, area, and shape index (LWR) based on the regression and structural breaks tests.

In the Discussion section, an index named frequency of structural changes (FSC) was created to identify the impact intensity of the TGD on the dynamics (including both how and where) of MCBs. Although it is widely accepted that the dam operation can affect the evolution of MCBs, the locations and magnitude of such effects are largely unknown. Based on the FSC, this study quantitatively analyzed the impacts of TGD operation on the MCBs dynamics (in both area and LWR change). Our quantitative results show MCBs area and LWR both decreased as the distance to the TGD increasing. Our results also identified the furthest locations of the TGD influences, i.e., DT for the LWR dynamics and the NJ for area dynamics (Fig. 14a). In addition to the location affected by TGD, this study analyzed the way of structural changes of MCBs. Our results show that the majority of structural changed MCBs experienced an opposite trend change in the post-TGD period compared to the pre-TGD period. Specifically, 72 out 140 MCBs showed PR structural change in area (positive in pre-TGD and negative in post-TGD) and 79 out 140 MCBs experienced NR structural change in LWR

(negative in pre-TGD and positive in post-TGD). These quantitative results indicate that the operation of the TGD could be the driving force that makes over half of MCBs experience an erosion condition (decreasing area) and become slim (increasing LWR) in the post-TGD period.

We didn't additionally analyze "when" would the MCBs be affected by TGD. According to our understanding, the most likely time for the TGD started to affect an MCB's evolution could be the close time of the TGD (that is 2003).

To better present the quantitative research methods and results, we have improved relevant descriptions in the manuscript. I sincerely hope the reviewer can reconsider the comment.

(2) In my personal perspective, the manuscript does not contain any relevant conclusion. The results are quite obvious, the methodology is not innovative and the conclusion irrelevant. It is not surprising to observe that small fluvial bars are the most likely to disappear after a flood. Therefore, I do not understand what is the innovative contribution of the present manuscript beside the construction of the dataset.

**Response:** Thank again for the comment. We agree with that "it is not surprising to observe small fluvial bars are the most likely to disappear after a flood". However, this statement is not a conclusion but it was mentioned in the Discussion Section 4.1. In fact, the main conclusions are as follows:

*"Most of the MCBs in terms of number (98 out of 140) and total area (1172 $km^2$ out of 1291 $km^2$) were scattered in the lower reach (HK-EST) with an average interval of 10 km along the channel. The temporal dynamics patterns were revealed with annual MCBs data using a statistical classification system. This classification system grouped the 140 MCBs into four size-types based on their area histogram distribution pattern: T1 small size (area < 2$km^2$) (50% of total number), T2 middle size (area 2 - 7 $km^2$) (25%), T3 large size (area 7 - 33 $km^2$) (20%), and T4 extra-large size (area > 33 $km^2$) (5%). For each type, the MCBs' temporal dynamics in total number, area, and shape index (i.e., LWR) were comparatively analyzed pre- and post- TGD operation periods.*

*Overall, the total MCBs number increased before TGD operation and then declined substantially after the TGD operation. Regarding the different MCB types, only the T1 MCBs experienced big change in numbers and most of them happened in the lower reach. Although the areas of all types of MCBs showed overall increase trends, large size MCBs tended to experience larger change rates and fewer variations than those of the small size MCBs. In addition, large size MCBs seemed to receive fewer impacts of TGD on their area dynamics whereas the small size MCBs likely to have more influences from the TGD operation. As for the shape dynamics, small size and middle size MCBs tended to become relatively shorter and wider whereas the large and extra-large size MCB tended to become slim. Similarly, the shape dynamics of the large MCBs were more stable than those of small ones. This study implies that more attention is needed for the scale (size) effects of MCBs on their temporal dynamics in the future MCBs' analysis and MCB management such as channel dredging.*

*The operation of TGD could have significant effects on MCBs dynamics. The study shows that the strength of such effects decreased as the distance to the TGD increased, and minimized at HK (LWR dynamics) or JJ (area dynamics). In contrast, the driving forces of the MCBs dynamics in the lowest JJ-EST reach were more complex as more external influences such as sand mining activities were observed in the area and more additional analyses were needed in the future."*

We have done a comprehensive search in Web of Science, we cannot find any similar studies in

the database. Please see the screenshot as follows:

[Figure]

As to innovations, this research made the first attempt to comprehensively estimate the longitudinal and temporal dynamics of MCBs during pre- and post-TGD periods, and their linkages to the TGD across the entire downstream reach based on quantitative data analysis.

(3)   The main contribution that may have some value is represented by the dataset reconstructed about the morphological evolution of the river system. Such a database may be relevant and useful, but I believe that the author should consider to change journal and eventually propose the manuscript for Earth System Science Data or Data in Brief. I do not think that the manuscript can be accepted in HESS with any attempt to provide a physical explanation of the dynamics of fluvial bar and of their mutual interaction with the dam management.

**Response:** We appreciate the reviewer's acknowledgment of the new contribution of our data. We also thank the reviewer for the kind suggestion for different journal selections. As we emphasized in the previous responses, this piece of work is the first study that   comprehensively estimated dynamics of MCBs and analyzed the influence the TGD on the spatiotemporal changes of MCBs. In addition to a new dataset, our study made a new contribution to understanding the impact of TGD on the MCBs. Therefore, HESS is the most suitable journal to disseminate our research.